# Faster Margin Maximization Rates for Generic Optimization Methods

**Guanghui Wang**[1], **Zihao Hu**[1], **Vidya Muthukumar**[2,3], **Jacob Abernethy**[1,4]

[1]College of Computing, Georgia Institute of Technology
[2]School of Electrical and Computer Engineering, Georgia Institute of Technology
[3]School of Industrial and Systems Engineering, Georgia Institute of Technology
[4]Google Research, Atalanta

{gwang369,zihaohu,vmuthukumar8}@gatech.edu, abernethyj@google.com

## Abstract

First-order optimization methods tend to inherently favor certain solutions over others when minimizing a given training objective with multiple local optima. This phenomenon, known as *implicit bias*, plays a critical role in understanding the generalization capabilities of optimization algorithms. Recent research has revealed that gradient-descent-based methods exhibit an implicit bias for the $\ell_2$-maximal margin classifier in the context of separable binary classification. In contrast, generic optimization methods, such as mirror descent and steepest descent, have been shown to converge to maximal margin classifiers defined by alternative geometries. However, while gradient-descent-based algorithms demonstrate fast implicit bias rates, the implicit bias rates of generic optimization methods have been relatively slow. To address this limitation, in this paper, we present a series of state-of-the-art implicit bias rates for mirror descent and steepest descent algorithms. Our primary technique involves transforming a generic optimization algorithm into an online learning dynamic that solves a regularized bilinear game, providing a unified framework for analyzing the implicit bias of various optimization methods. The accelerated rates are derived leveraging the regret bounds of online learning algorithms within this game framework.

## 1 Introduction

The training objective in the optimization of modern over-parametrized ML models typically presents various local optima with low training error. Despite this, empirical studies have demonstrated that first-order optimization methods generally converge to the solution with strong generalization, even without any explicit regularization (Neyshabur et al., 2014; Zhang et al., 2021). This observation has spurred interest in the study of the *implicit bias* of the algorithm. In other words, *among all potential parameter choices with low training error, which ones are inherently favored by optimization methods?*

For the classical linear classification problem with separable data, the pioneering works (Soudry et al., 2018; Ji & Telgarsky, 2018) reveal that minimizing the (unregularized) empirical risk with exponential loss by the classical gradient descent (GD) automatically *maximizes the $\|\cdot\|_2$-margin*, meaning its margin converges to that of the best classifier within an $\ell_2$-norm ball (referred to as the $\|\cdot\|_2$-maximal margin classifier). This finding implies that GD exhibits an *implicit bias* towards the $\|\cdot\|_2$-maximal margin classifier, which helps account for its favorable generalization performance. However, these works show that GD only maximizes the $\|\cdot\|_2$-margin at a slow $O\left(\frac{\log n}{\log T}\right)$ rate, where $T$ is the time horizon, and $n$ is the cardinality of the data set. Since then, several *faster* margin

maximization rates have been reported. Nacson et al. (2019) revealed that for the exponential loss, GD with an aggressive step size attains an $O\left(\frac{\log n + \log T}{\sqrt{T}}\right) \|\cdot\|_2$-margin maximization rate. This result was later improved to $O\left(\frac{\log n}{T}\right)$ by Ji & Telgarsky (2021), via an elegant primal-dual analysis. Recent work (Ji et al., 2021; Wang et al., 2022b) further proved that momentum-based GD and Nesterov-accelerated GD can maximize the $\|\cdot\|_2$-margin at an $O\left(\frac{\log n}{T^2}\right)$ rate.

As the implicit bias of gradient-descent-based methods becomes better understood, it is natural to explore similar characterizations for other optimization methods. Note that, since gradient-descent-based methods are biased towards the $\|\cdot\|_2$-maximal margin classifier, they might generalize poorly when the data does not adhere to the $\ell_2$-geometry, as shown in Gentile (2000); Chen et al. (2001). This limitation suggests that it is essential to study the implicit bias of alternative optimization methods that could potentially be biased in different directions. Two such methods include steepest descent with respect to different norms and mirror descent with different potentials. For instance, Gunasekar et al. (2018a) demonstrate that for the exponential loss, the steepest descent algorithm with respect to a general norm $\|\cdot\|$ asymptotically converges to the corresponding $\|\cdot\|$-maximal margin classifier. This result implies that the steepest descent algorithm can adapt to different data geometries by changing the norm used in the algorithm. On the other hand, Sun et al. (2022) show that the mirror descent algorithm with the potential $\|\cdot\|_q^q$ for $q > 1$ can maximize the $\|\cdot\|_q$-margin at a rate of $O\left(\frac{1}{(\log T)^{q-1}}\right)$.

While the asymptotic directional convergence of these generic optimization methods is well-understood, a natural question remains: *can generic optimization methods (e.g., mirror descent and steepest descent) achieve faster margin maximization rates?* Several papers have contributed partial answers to this question. Li et al. (2021) show that mirror descent with an aggressive step size maximizes the margin in an $O\left(\frac{\log n}{T^{1/4}}\right)$ rate. However, their analysis assumes the potential function is *both strongly-convex and smooth* with respect to some norm, and thus is mainly limited to the $\ell_2$-geometry. For steepest descent with respect to a general norm $\|\cdot\|$, Nacson et al. (2019) prove that an $O\left(\frac{\log n + \log T}{\sqrt{T}}\right) \|\cdot\|$-margin maximization rate can be achieved with an appropriately chosen step size, but it is unclear whether it can be further improved. In this paper, we provide the fastest known rates for margin maximization for generic optimization methods, through the following contributions:

- First, we study a *weighted-average* version of mirror descent with the squared $\ell_q$-norm $\frac{1}{2}\|\cdot\|_q^2$ as the potential for $q \in (1, 2]$. This potential function is strongly convex but not smooth. We show that, with an appropriately chosen step size, the algorithm achieves a faster $\|\cdot\|_q$-margin maximization rate on the order of $O\left(\frac{\log n \log T}{(q-1)T}\right)$. We also further improve the rate to $O\left(\frac{1}{T(q-1)} + \frac{\log n \log T}{T^2}\right)$ with a more aggressive step size. When $q = 2$, the algorithm reduces to average GD, and our rate $O\left(\frac{1}{T} + \frac{\log n \log T}{T^2}\right)$ is a $\log n$-factor tighter than the $O\left(\frac{\log n}{T}\right)$ rate of the last-iterate GD (Ji & Telgarsky, 2021).

- Next, for the steepest descent algorithm with respect to the $\ell_q$-norm for $q \in (1, 2]$, we show the margin maximization rate can be improved from $O\left(\frac{\log n + \log T}{\sqrt{T}}\right)$ to $O\left(\frac{\log n}{T(q-1)}\right)$.

- Finally, we demonstrate that a even faster $O\left(\frac{\log n}{T^2(q-1)}\right) \|\cdot\|_q$-margin maximization rate can be achieved in two ways: a) mirror descent with Nesterov acceleration, or b) steepest descent with extra gradient and momentum.

The essential premise for our approach is that *minimizing empirical risk (ERM) with generic optimization methods can be equivalently viewed as solving a regularized bilinear game with online learning dynamics*. Within this framework, we design new pairs of online learning methods whose outputs (and, by extension, the outputs of the corresponding generic optimization methods) automatically maximize the margin. The convergence rates are determined by the time-averaged regret bounds of these online learning algorithms when played against each other, which turn out to be much faster than the worst-case $O(1/\sqrt{T})$ rate.

Wang et al. (2022b) were the first to draw parallels between *Nesterov-accelerated GD* for ERM and solving the bilinear game through an online dynamic. However, it was still open that whether this kind of analysis suits other GD-based methods. Moreover, the non-linearity of the mirror map in generic optimization methods makes analysis particularly challenging. In this paper, we reveal that the game framework can in fact encompass implicit bias analysis for a range of generic optimization methods, and offer a more streamlined and unified analysis. Within this game framework, we derive several other results beyond those mentioned above:

- By selecting suitable online learning algorithms, we obtain a momentum-based data-dependent MD algorithm with an $O(\frac{\mathcal{V}_T}{T^2(q-1)} + \frac{\log n \log T}{T^2}) \| \cdot \|_q$-margin maximization rate, where $\mathcal{V}_T = \sum_{t=2}^{T} \|\mathbf{p}_t - \mathbf{p}_{t-1}\|_1^2$ is the path-length of a series of distributions on the training data $\mathbf{p}_t$. In the worst case, this reduces to the margin maximization rate of MD, but this can be much tighter if $\mathcal{V}_T$ is sublinear in $T$.

- Apart from margin maximization rates, we also demonstrate the corresponding directional error, i.e., the bound on the $\ell_q$-distance between the maximal margin classifier and the normalized output of the generic methods, which are also controlled by the regret bounds of two-players against each other. This kind of convergence rates are new for most of the generic methods. In general, we prove that the directional errors are typically a square-root factor worse than the margin maximization rates.

- For our steepest descent, by setting the norm to the general norm $\| \cdot \|$ and the $\ell_2$-norm respectively, we can recover the algorithms and theoretical guarantees in Nacson et al. (2019); Ji & Telgarsky (2021) under the game framework. This implies that these algorithms can also be viewed as solving a *regularized* bilinear game using online learning algorithms, offering a deeper understanding of the role of implicit bias in optimization methods.

**Additional related work**    The strategy of solving a zero-sum game using online learning algorithms playing against each other has been extensively studied, primarily through the lens of *independent learning agents* (e.g., Rakhlin & Sridharan, 2013; Daskalakis et al., 2018; Wang & Abernethy, 2018; Daskalakis & Panageas, 2019; Zhang et al., 2022). In contrast, our central motivation and challenge lies in identifying the exact equivalent forms of generic optimization algorithms under the regularized bilinear game dynamic. Our framework is also motivated by the line of research that employs the *Fenchel-game* to elucidate commonly used convex optimization methods (Abernethy et al., 2018; Wang & Abernethy, 2018; Wang et al., 2021b). However, our framework diverges significantly from these approaches. These works focus on the convergence of the optimization problem itself, while our framework emphasizes that the choice of optimization algorithm, which solely targets the minimization of empirical risk, has a significant impact on maximizing the margin, which we might view as an "algorithmic externality." Max-margin guarantees can not arise from convergence of the ERM objective alone, as there are typically multiple global minima in ERM minimization. Our analysis also considers an entirely different min-max problem than that of the Fenchel game (Wang et al., 2021b). Consequently, the correspondences we establish between optimization algorithms and online dynamics also differ. Finally, we note that previous work has also analyzed the implicit bias through direct primal optimization analysis (e.g., Nacson et al., 2019; Sun et al., 2022) or using a dual perspective (e.g., Ji & Telgarsky, 2021; Ji et al., 2021). For the former analysis, it is unclear whether and how faster rates can be obtained. For the latter, it remains an open question how to extend the framework beyond $\ell_2$-geometry, which in some sense was the motivation for the present work. For more related work, we refer the reader to Appendix A.

## 2 Preliminaries

In this section, we present our basic setting along with some standard assumptions and definitions.

**Notation**    We use lower case bold face letters $\mathbf{x}, \mathbf{y}$ to denote vectors, lower case letters $a, b$ to denote scalars, and upper case bold face letters $\mathbf{A}, \mathbf{B}$ to denote matrices. For a vector $\mathbf{x} \in \mathbb{R}^d$, we use $x_i$ to denote the $i$-th component of $\mathbf{x}$. For a matrix $\mathbf{A} \in \mathbb{R}^{n \times d}$, let $\mathbf{A}_{(i,:)}$ be its $i$-th row, $\mathbf{A}_{(:,j)}$ the $j$-th column, and $\mathbf{A}_{(i,j)}$ the $i$-th element of the $j$-th column. $\forall \mathbf{x} \in \mathbb{R}^d$, we use $\| \cdot \|$ to denote a general norm in $\mathbb{R}^d$, $\| \cdot \|_*$ its dual norm, $\|\mathbf{x}\|_q$ the $q$-norm of $\mathbf{x}$, defined as $\|\mathbf{x}\|_q = (\sum_{i=1}^{d} |x_i|^q)^{1/q}$, and $q \in (1, 2]$. We use $\| \cdot \|_p$ to denote the dual norm of $q$-norm, where $p \in [2, \infty)$, $\frac{1}{p} + \frac{1}{q} = 1$.

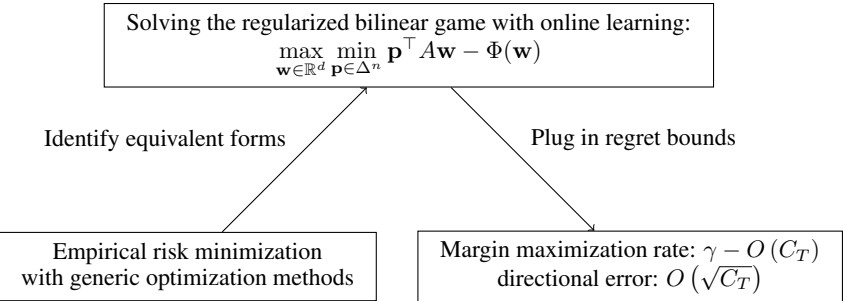

Figure 1: Illustration of the game framework for implicit bias analysis. In Section 3, we show that solving a regularized bilinear game with online learning algorithms (top box) can directly maximize the margin, and the convergence rate is on the same order of the averaged regret $C_T$ (right box); In Sections 4, we prove that minimizing the empirical risk with a series of generic optimization methods (left box) is equivalent to using online learning algorithms to solve the regularized bilinear game. Thus, the implicit bias rates can be directly obtained by plugging in the regret bounds.

We denote $\mathcal{B}_{\|\cdot\|}$ the $\|\cdot\|$-ball, defined as $\mathcal{B}_{\|\cdot\|} = \{\mathbf{x} \in \mathbb{R}^d | \|\mathbf{x}\| \leq 1\}$. $\forall \mathbf{x}, \mathbf{x}' \in \mathbb{R}^d$, we define the Bregman divergence between $\mathbf{x}$ and $\mathbf{x}'$ with respect to a strictly convex potential function $\Phi(\mathbf{x})$ as $D_\Phi(\mathbf{x}, \mathbf{x}') = \Phi(\mathbf{x}) - \Phi(\mathbf{x}') - \nabla\Phi(\mathbf{x}')^\top(\mathbf{x} - \mathbf{x}')$. For a positive integer $n$, we denote $\{1, \ldots, n\}$ as $[n]$, and the $(n-1)$-dimensional simplex as $\Delta^n$. Let $E : \Delta^n \mapsto \mathbb{R}$ be the negative entropy function, defined as $E(\mathbf{p}) = \sum_{i=1}^n p_i \log p_i, \forall \mathbf{p} \in \Delta^n$.

**Basic setting**  Consider a set of $n$ data points $\mathcal{S} = \{(\mathbf{x}^{(i)}, y^{(i)})\}$, where $\mathbf{x}^{(i)} \in \mathbb{R}^d$ is the feature vector for the $i$-th example, and $y^{(i)} \in \{-1, +1\}$ the corresponding binary label. We are interested the optimization trajectory of first-order methods for minimizing the following unbounded and unregularized empirical risk:

$$\min_{\mathbf{w} \in \mathbb{R}^d} L(\mathbf{w}) = \frac{1}{n} \sum_{i=1}^n r(\mathbf{w}^\top \mathbf{x}^{(i)}; y^{(i)}), \tag{1}$$

where $\mathbf{w} \in \mathbb{R}^d$ is a linear classifier, $r : \mathbb{R} \times \{\pm 1\} \mapsto \mathbb{R}$ is the loss function. Following previous work, we focus on the exponential loss, given by $r(\mathbf{w}^\top \mathbf{x}; y) = \exp(-y\mathbf{x}^\top \mathbf{w})$. We introduce the following standard assumption and definitions.

**Definition 1** ($\|\cdot\|$-margin). *For a linear classifier $\mathbf{w} \in \mathbb{R}^d$ and a norm $\|\cdot\|$, we define its $\|\cdot\|$-margin as*

$$\widetilde{\gamma}(\mathbf{w}) = \frac{\min\limits_{i \in [n]} y^{(i)}\mathbf{w}^\top \mathbf{x}^{(i)}}{\|\mathbf{w}\|} = \frac{\min\limits_{\mathbf{p} \in \Delta^n} \mathbf{p}^\top \mathbf{A}\mathbf{w}}{\|\mathbf{w}\|},$$

*where $\mathbf{A} = [\ldots; y^{(i)}\mathbf{x}^{(i)\top}; \ldots] \in \mathbb{R}^{n \times d}$ is the matrix that contains all data.*

**Assumption 1.** *Assume $\mathcal{S}$ is linearly separable and bounded with respect to some norm $\|\cdot\|$. More specifically, we assume $\exists \mathbf{w}^*_{\|\cdot\|} \in \mathcal{B}_{\|\cdot\|}$, s.t., $\mathbf{w}^*_{\|\cdot\|} = \arg\max_{\|\mathbf{w}\| \leq 1} \min_{i \in [n]} y^{(i)}\mathbf{x}^{(i)\top}\mathbf{w}$, whose margin $\widetilde{\gamma}(\mathbf{w}^*_{\|\cdot\|}) = \gamma > 0$. We refer to $\mathbf{w}^*_{\|\cdot\|}$ as the $\|\cdot\|$-maximal margin classifier. Note that, for any $\mathbf{w} \in \mathbb{R}^d$, if $\widetilde{\gamma}(\mathbf{w}) = \gamma$, $\mathbf{w}$ and $\mathbf{w}^*_{\|\cdot\|}$ are at the same direction. Finally, we also assume $\mathcal{S}$ is bounded wrt some dual norm $\|\cdot\|_*$, i.e., $\forall i \in [n]$, $\|\mathbf{x}^{(i)}\|_* \leq 1$.*

**Definition 2** ($\|\cdot\|$-Margin maximization rate and $\|\cdot\|$-directional error). *Suppose Assumption 1 is satisfied. We consider a sequence of solutions $\mathbf{w}_1, \ldots, \mathbf{w}_t, \ldots$, and state that $\mathbf{w}_t$ converges to $\mathbf{w}^*_{\|\cdot\|}$ if either $\lim_{t \to \infty} \widetilde{\gamma}(\mathbf{w}_t) \to \gamma$, or $\lim_{t \to \infty} \|\frac{\mathbf{w}_t}{\|\mathbf{w}_t\|} - \mathbf{w}^*_{\|\cdot\|}\| \to 0$. We define the upper bound on $|\gamma - \widetilde{\gamma}(\mathbf{w}_t)|$ the $\|\cdot\|$-margin maximization rate, and $\|\frac{\mathbf{w}_t}{\|\mathbf{w}_t\|} - \mathbf{w}^*_{\|\cdot\|}\|$ the $\|\cdot\|$-directional error.*

## 3  A Game Framework for Maximizing the Margin

In this section, we present a general game framework and demonstrate that solving this game with online learning algorithms can directly maximize the margin and minimize the directional error.

**Protocol 1** No-regret dynamics with weighted OCO for solving $g(\mathbf{p}, \mathbf{w})$

---

1: **Initialization**: $\mathsf{OL^w}, \mathsf{OL^P}$. // The online algorithms for choosing $\mathbf{w}$ and $\mathbf{p}$.
2: **for** $t = 1, \ldots, T$ **do**
3: $\quad \mathbf{w}_t \leftarrow \mathsf{OL^w}$;
4: $\quad \mathsf{OL^P} \leftarrow \alpha_t, \ell_t(\cdot)$; // Define $\ell_t(\cdot) = g(\mathbf{w}_t, \cdot)$
5: $\quad \mathbf{p}_t \leftarrow \mathsf{OL^P}$;
6: $\quad \mathsf{OL^w} \leftarrow \alpha_t, h_t(\cdot)$; // Define $h_t(\cdot) = -g(\cdot, \mathbf{p}_t)$
7: **end for**
8: **Output**: $\widetilde{\mathbf{w}}_T = \sum_{t=1}^{T} \alpha_t \mathbf{w}_t$.

---

Then, in Section 4, we show that many generic optimization methods can be considered as solving this game with different online dynamics. As a result, the margin maximization rate (and also the directional error) of these optimization methods are exactly characterized by the regret bounds of the corresponding online learning algorithms. We illustrate this procedure in Figure 1. The game objective is defined as follows:

$$\max_{\mathbf{w} \in \mathbb{R}^d} \min_{\mathbf{p} \in \Delta^n} g(\mathbf{p}, \mathbf{w}) = \mathbf{p}^\top \mathbf{A} \mathbf{w} - \Phi(\mathbf{w}), \tag{2}$$

where $\Phi(\mathbf{w}) = \frac{1}{2}\|\mathbf{w}\|^2$ is a regularizer and $\|\cdot\|$ denotes some *general norm* in $\mathbb{R}^d$. Following previous work (Wang et al., 2021b, 2022b), we apply a weighted no-regret dynamic protocol (summarized in Protocol 1) to solve the game. We first give a brief introduction of Protocol 1, and then present the theorem about the margin of its output.

**Description of Protocol 1**  In Protocol 1, the players of the zero-sum game try to find the equilibrium by applying online learning algorithms. In each round $t$, the $\mathbf{p}$-player first picks a decision $\mathbf{p}_t$, and passes a weighted loss function to the $\mathbf{w}$-player, defined as

$$\alpha_t h_t(\mathbf{w}) = -\alpha_t(\mathbf{p}_t^\top \mathbf{A} \mathbf{w} - \Phi(\mathbf{w})) = -\alpha_t g(\mathbf{p}_t, \mathbf{w}).$$

Then, the $\mathbf{w}$-player observes the loss, picks a decision $\mathbf{w}_t$, and passes a weighted loss function

$$\alpha_t \ell_t(\mathbf{p}) = \alpha_t(\mathbf{p}^\top \mathbf{A} \mathbf{w}_t - \Phi(\mathbf{w}_t)) = \alpha_t g(\mathbf{p}, \mathbf{w}_t),$$

to the $\mathbf{p}$-player. Note that the order of the two players can also be reversed. After $T$ iterations, the algorithm outputs the weighted sum of the $\mathbf{w}$-player's decisions: $\widetilde{\mathbf{w}}_T = \sum_{t=1}^{T} \alpha_t \mathbf{w}_t$. Under this framework, we define the weighted regret upper bound of both players respectively as

$$\sum_{t=1}^{T} \alpha_t \ell_t(\mathbf{p}_t) - \min_{\mathbf{p} \in \Delta^n} \sum_{t=1}^{T} \alpha_t \ell_t(\mathbf{p}) \le \mathrm{Reg}_T^{\mathbf{P}}, \quad \text{and} \quad \sum_{t=1}^{T} \alpha_t h_t(\mathbf{w}_t) - \min_{\mathbf{w} \in \mathbb{R}^d} \sum_{t=1}^{T} \alpha_t h_t(\mathbf{w}) \le \mathrm{Reg}_T^{\mathbf{w}}.$$

Denote the upper bound on the *average* weighted regret by $C_T = (\mathrm{Reg}_T^{\mathbf{P}} + \mathrm{Reg}_T^{\mathbf{w}}) / \sum_{t=1}^{T} \alpha_t$. We have the following conclusion on the margin and directional error of $\widetilde{\mathbf{w}}_T$. The proof of this theorem can be found in Appendix B.

**Theorem 1.** *Suppose Assumption 1 holds with respect to some general norm $\|\cdot\|$. Consider solving the two-player zero-sum game defined in* (2) *by applying Protocol 1. Then $\widetilde{\mathbf{w}}_T$ will have a positive margin on round $T$ if $C_T \le \frac{\gamma^2}{4}$. Moreover, as long as $C_T \le \frac{\gamma^2}{4}$, we have*

$$\frac{\min_{\mathbf{p} \in \Delta^n} \mathbf{p}^\top \mathbf{A} \widetilde{\mathbf{w}}_T}{\|\widetilde{\mathbf{w}}_T\|} \ge \gamma - \frac{4C_T}{\gamma^2}. \tag{3}$$

*If $\Phi(\mathbf{w})$ is $\lambda$-strongly convex with wrt $\|\cdot\|$, we have*

$$\left\| \frac{\widetilde{\mathbf{w}}_T}{\|\widetilde{\mathbf{w}}_T\|} - \mathbf{w}_{\|\cdot\|}^* \right\| \le \frac{8\sqrt{2}}{\gamma^2 \sqrt{\lambda}} \sqrt{C_T}.$$

Theorem 1 shows that the output of Protocol 1, denoted as $\widetilde{\mathbf{w}}_T$, achieves a positive margin when the average regret $C_T \le \frac{\gamma^2}{4}$. In the following sections, we demonstrate that with appropriately chosen

online learning algorithms $C_T$ always decreases with respect to $T$; in fact $C_T \to 0$ as $T \to \infty$. Therefore, once the condition $C_T \leq \frac{\gamma^2}{4}$ is met for a particular value $T_0$, it will also be met for all $T \geq T_0$. Thereafter, $\widetilde{\mathbf{w}}_T$ continues to increase the $\|\cdot\|$-margin and converges to the maximum $\|\cdot\|$-margin classifier, and the rate is directly characterized by $C_T$. Since $C_T$ is the average regret of the online learning algorithms, better bounds on $C_T$ lead to a less stringent condition on large enough $T$. Finally, we note that the condition on sufficiently large $T$ is also (explicitly or implicitly) required in all previous work on the non-asymptotic margin maximization rates of generic methods (Nacson et al., 2019; Li et al., 2021; Sun et al., 2022). We refer to Appendix A for more details.

## 4   Implicit Bias of Generic Methods

In this section, we show that average mirror descent and steepest descent can find their equivalent online learning forms under Protocol 1. Thus, their margin maximization rates can be directly characterized by the corresponding average regret $C_T$. For clarity, we use $\mathbf{v}_t$ to denote the classifier updates in the original methods, and $\mathbf{w}_t$ the update under the game framework. Note that Theorem 1 clearly implies that the convergence rate of the directional error is always a square-root worse than that of the margin maximization rate. Due to space limitations, we only present the margin maximization rates, while the corresponding rates on directional error are presented in the appendix.

### 4.1   Mirror-Descent-Type of Methods

First, we consider minimizing (1) by applying the following mirror descent algorithm:

$$\mathbf{v}_t = \operatorname*{argmin}_{\mathbf{v} \in \mathbb{R}^d} \eta_t \left\langle \nabla L(\mathbf{v}_{t-1}), \mathbf{v} \right\rangle + D_\Phi(\mathbf{v}, \mathbf{v}_{t-1}), \tag{4}$$

where $D_\Phi(\mathbf{v}, \mathbf{v}_{t-1})$ is the Bregman divergence between $\mathbf{v}$ and $\mathbf{v}_{t-1}$, and $\Phi(\mathbf{v})$ is a strongly convex potential function that defines the mirror map. Note that since the feasible domain in (4) is unbounded, we can rewrite the algorithm in the following form:

$$\nabla \Phi(\mathbf{v}_t) = \nabla \Phi(\mathbf{v}_{t-1}) - \eta_t \nabla L(\mathbf{v}_{t-1}).$$

In this paper, we consider weighted-average mirror descent with the squared $q$-norm, i.e., $\Phi(\mathbf{w}) = \frac{1}{2}\|\mathbf{w}\|_q^2$, where $q \in (1, 2]$, and demonstrate that this optimization algorithm can enable faster $\|\cdot\|$-margin maximization rates. The detailed update rule is summarized in the left box of Algorithm 1. It is worth noting that this type of regularizer is $(q-1)$-strongly convex with respect to $\|\cdot\|_q$, and can be updated efficiently in closed form as below[1]: for each coordinate $i \in [d]$, we have

$$
\begin{aligned}
\widehat{v}_{t,i} &= \operatorname{sign}(v_{t-1,i})|v_{t-1,i}|^{q-1}\|\mathbf{v}_{t-1}\|_q^{2-q} - \eta_t[\nabla L(\mathbf{v}_{t-1})]_i, \\
v_{t,i} &= \operatorname{sign}(\widehat{v}_{t,i})|\widehat{v}_{t,i}|^{p-1}\|\widehat{\mathbf{v}}_t\|_p^{2-p}.
\end{aligned}
\tag{5}
$$

We make a few final observations about this algorithm: 1) Instead of using the weighted sum $\widetilde{\mathbf{v}}_T$, we could output the weighted average $\frac{\widetilde{\mathbf{v}}_t}{\sum_{s=1}^t \alpha_s}$ without altering the margin or directional convergence rate. This is attributed to the scale-invariance of the margin, i.e., $\forall c > 0, \mathbf{w} \in \mathbb{R}^d, \widetilde{\gamma}(\mathbf{w}) = \widetilde{\gamma}(c\mathbf{w})$. The same argument applies to the directional error. 2) The use of the weighted average is standard in the analysis of mirror descent (e.g., Section 4.2 of Bubeck et al. (2015)). This paper shows that using non-uniform weights is advantageous for achieving rapid margin maximization rates; 3) The per-round computational complexity of (5) is $O(d)$, which is similar to that of $p$-mirror-descent of Sun et al. (2022). However, we note that the $p$-MD algorithm of Sun et al. (2022) does not need to compute the norm of the decision at each round, which could be more efficient in real-world applications where parallel or distributed computation is desired.

For Algorithm 1, we have the following theorem. We present its proof in Appendix C, along with a more general theorem that allows a general configuration of the parameters $\eta_t$, $\alpha_t$ and $\beta_t$.

**Theorem 2.** *Suppose Assumption 1 holds wrt $\|\cdot\|_q$-norm for $q \in (1, 2]$. For the left box of Algorithm 1, let $\eta_t = \frac{1}{L(\mathbf{v}_{t-1})}$. For the right box, let $\alpha_t = 1$, and $\beta_1 = 1$, $\beta_t = \frac{1}{t-1}$. Then the methods in the two boxes of Algorithm 1 are identical, in the sense that $\widetilde{\mathbf{v}}_T = \widetilde{\mathbf{w}}_T$. Moreover, we have the average*

---

[1]This expression appears in (Section 6.7, Orabona, 2019) and we reproduce it for completeness.

**Algorithm 1** Mirror Descent [Recall $\ell_t(\mathbf{p}) = g(\mathbf{p}, \mathbf{w}_t)$, and $h_t(\mathbf{w}) = -g(\mathbf{p}_t, \mathbf{w})$]

| | |
|---|---|
| 1: **for** $t = 1, \ldots, T$ **do** | **p-player:** $\mathbf{p}_t = \underset{\mathbf{p} \in \Delta^n}{\mathrm{argmin}} \ \alpha_t \ell_{t-1}(\mathbf{p}) + \beta_t D_{\mathrm{KL}}\left(\mathbf{p}, \frac{1}{n}\right)$ |
| 2: $\quad \nabla \Phi(\mathbf{v}_t) = \nabla \Phi(\mathbf{v}_{t-1}) - \eta_t \nabla L(\mathbf{v}_{t-1})$ | **w-player:** $\mathbf{w}_t = \underset{\mathbf{w} \in \mathbb{R}^d}{\mathrm{argmin}} \sum_{j=1}^{t} \alpha_j h_j(\mathbf{w})$ |
| 3: **end for** | |
| 4: **Output:** $\widetilde{\mathbf{v}}_T = \sum_{t=1}^{T} \frac{1}{t} \mathbf{v}_t$ | **Output:** $\widetilde{\mathbf{w}}_T = \sum_{t=1}^{T} \alpha_t \mathbf{w}_t$ |

*regret upper bound* $C_T = \frac{\left(\frac{2}{q-1} + 2\log n\right)(\log T + 2)}{T}$. *Therefore, the algorithm achieves a positive margin when $T$ is sufficiently large such that* $T \geq \frac{4\left(\frac{2}{q-1} + 2\log n\right)(\log T + 2)}{\gamma^2}$. *We have the convergence rate*

$$\frac{\min_{\mathbf{p} \in \Delta^n} \mathbf{p}^\top \mathbf{A} \widetilde{\mathbf{v}}_T}{\|\widetilde{\mathbf{v}}_T\|_q} \geq \gamma - \frac{4\left(\frac{2}{q-1} + 2\log n\right)(\log T + 2)}{\gamma^2 T} = \gamma - O\left(\frac{\log n \log T}{(q-1)\gamma^2 T}\right), \qquad (6)$$

*and*

$$\left\| \frac{\widetilde{\mathbf{v}}_T}{\|\widetilde{\mathbf{v}}_T\|_q} - \mathbf{w}^*_{\|\cdot\|_q} \right\|_q \leq \frac{8\sqrt{2}}{\gamma^2 \sqrt{(q-1)}} \sqrt{\frac{\left(\frac{2}{q-1} + 2\log n\right)(\log T + 2)}{T}} = O\left(\frac{\sqrt{\log n \log T}}{\gamma^2(q-1)\sqrt{T}}\right).$$

The first part of Theorem 2 indicates that the mirror descent algorithm can be described as two players using certain cleverly designed online learning algorithms to solve the *regularized* bilinear game in (2). More specifically, for the **p**-player, we propose a new and unusual online learning algorithm, which we call *regularized greedy*.

$$\mathbf{p}_t = \underset{\mathbf{p} \in \Delta^n}{\mathrm{argmin}} \ \alpha_t \ell_{t-1}(\mathbf{p}) + \beta_t D_{\mathrm{KL}}\left(\mathbf{p}, \frac{1}{n}\right).$$

Essentially, in round $t$, the **p**-player minimizes the previous round's loss function, $\ell_{t-1}$, plus a regularizer at round $t$, and the two terms are balanced by the parameters $\alpha_t$ and $\beta_t$. On the other hand, we select the *follow-the-leader$^+$* algorithm for the **w**-player:

$$\mathbf{w}_t = \underset{\mathbf{w} \in \mathbb{R}^d}{\mathrm{argmin}} \sum_{j=1}^{t} \alpha_j h_j(\mathbf{w}),$$

which returns the solution that minimize the cumulative loss so far. The $+$ sign in the name is because the algorithm can pick the decision $\mathbf{w}_t$ *after* seeing its loss function. This is an interesting and unusual design because the regularized greedy algorithm will clearly suffer a worst-case *linear* regret for the **p**-player. Fortunately, for our specific case we are able to prove a sharper *data-dependent* regret bound for the **p**-player as below:

$$\mathrm{Reg}_T^{\mathbf{p}} = O\left( \sum_{t=2}^{T} \frac{(t-1)(q-1)}{2} \|\mathbf{w}_t - \mathbf{w}_{t-1}\|_q^2 + \log n \log T \right),$$

Therefore, the dominating term (i.e., the first term above) of the **p**-player's regret bound can be canceled by the **w**-player's regret bound, given by:

$$\mathrm{Reg}_T^{\mathbf{w}} = O\left( -\sum_{t=2}^{T} \frac{(t-1)(q-1)}{2} \|\mathbf{w}_t - \mathbf{w}_{t-1}\|_q^2 \right).$$

Note that the **w**-player's regret bound is negative as the corresponding algorithm used is *clairvoyant*, i.e. can see the current loss $\ell_t$ before making a decision at round $t$. This ensures that sublinear (and more generally fast) rates are possible. Note that $\beta_t$ and $\alpha_t$ will influence both the regret bound and the algorithm equivalence analysis, so finding the right parameter configuration that works for both is a non-trivial task. We make the choice $\beta_t = \frac{\alpha_t}{\sum_{i=1}^{t-1} \alpha_i}$, which ensures both algorithmic equivalence and sublinear regret bounds.

**Algorithm 2** Momentum-based MD [Recall $\ell_t(\mathbf{p}) = g(\mathbf{p}, \mathbf{w}_t)$, and $h_t(\mathbf{w}) = -g(\mathbf{p}_t, \mathbf{w})$]

| | |
|---|---|
| 1: **for** $t = 1, \ldots, T$ **do** | **w-player:** |
| 2: $\quad \nabla\Phi(\mathbf{v}_t) = \nabla\Phi(\mathbf{v}_{t-1}) - \eta_t \nabla L(\mathbf{v}_{t-1})$ | $\mathbf{w}_t = \text{argmin}_{\mathbf{w} \in \mathbb{R}^d} \sum_{i=1}^{t-1} \alpha_i h_i(\mathbf{w}) + \alpha_t h_{t-1}(\mathbf{w})$ |
| $\quad - (\widehat{\eta}_t \nabla L(\mathbf{v}_{t-1}) - \eta_{t-1} \nabla L(\mathbf{v}_{t-2}))$ | **p-player:** |
| 3: **end for** | $\mathbf{p}_t = \text{argmin}_{\mathbf{p} \in \Delta^n} \alpha_t \ell_t(\mathbf{p}) + \beta_t D_{\text{KL}}(\mathbf{p}, \mathbf{p}_0)$ |
| 4: **Output:** $\widetilde{\mathbf{v}}_T = \sum_{t=1}^{T} \frac{1}{t} \mathbf{v}_t$ | **Output:** $\widetilde{\mathbf{w}}_T = \sum_{t=1}^{T} \alpha_t \mathbf{w}_t$ |

The second part of Theorem 2 shows that the average regret $C_T$ of Algorithm 1 is on the order of $O\left(\frac{\log n \log T}{(q-1)\gamma^2 T}\right)$. Therefore, by plugging in Theorem 1, we observe that the margin shrinks on the order of $\gamma - O\left(\frac{\log n \log T}{\gamma^2(q-1)T}\right)$, and the implicit bias convergence rate is $O\left(\frac{\log n \log T}{\gamma^2(q-1)\sqrt{T}}\right)$. Next, we show an improved rate with a more aggressive step size on the order of $O\left(\frac{t}{L(\mathbf{v}_t)}\right)$ instead of $O\left(\frac{1}{L(\mathbf{v}_t)}\right)$. The proof of this result is given in Appendix C.

**Theorem 3.** *Suppose Assumption 1 holds wrt $\|\cdot\|_q$-norm for $q \in (1, 2]$. For the left box of Algorithm 1, let $\eta_t = \frac{t}{L(\mathbf{v}_{t-1})}$, and let the final output be $\widetilde{\mathbf{v}}_T = \sum_{t=1}^{T} \frac{2}{t+1} \mathbf{v}_t$. For the right box, let $\alpha_t = t$, and $\beta_1 = 1$, $\beta_t = \frac{2}{t-1}$. Then the two algorithms are identical, in the sense that $\widetilde{\mathbf{v}}_T = \widetilde{\mathbf{w}}_T$. Moreover, when $T \geq \sqrt{\frac{8\left[\frac{4T}{q-1} + 4\log n \log T + 1 + 2\log n\right]}{\gamma^2}}$, we have*

$$\frac{\min_{\mathbf{p} \in \Delta^n} \mathbf{p}^\top \mathbf{A} \widetilde{\mathbf{v}}_T}{\|\widetilde{\mathbf{v}}_T\|_q} \geq \gamma - \frac{32}{\gamma^2 T(q-1)} - \frac{8(4\log n \log T + 1 + 2\log n)}{\gamma^2 T^2}, \tag{7}$$

*and*

$$\left\| \frac{\widetilde{\mathbf{v}}_T}{\|\widetilde{\mathbf{v}}_T\|_q} - \mathbf{w}^*_{\|\cdot\|_q} \right\|_q \leq \frac{8\sqrt{2}}{\gamma^2\sqrt{q-1}} \sqrt{\frac{8}{(q-1)T} + \frac{4\log n \log T + 2\log n + 1}{T^2}}.$$

Observe that the margin maximization rate in Theorem 3 is $O\left(\frac{1}{(q-1)\gamma^2 T}\right) + O\left(\frac{\log n \log T}{\gamma^2 T^2}\right)$. Compared to (6), it has a better dependence on $\log n$ and $\log T$.

Finally, we focus on a momentum-based mirror descent algorithm, which is given in Algorithm 2. For this algorithm, we have the following guarantee.

**Theorem 4.** *Suppose Assumption 1 holds wrt $\|\cdot\|_q$-norm for $q \in (1, 2]$. For the left box of Algorithm 2, let $\eta_t = \frac{t}{L(\mathbf{v}_{t-1})}$, and $\widehat{\eta}_t = \frac{t-1}{L(\mathbf{v}_{t-1})}$. For the second box, let $\alpha_t = t$, and $\beta_t = \frac{2}{t+1}$. Then the methods in the two boxes of Algorithm 2 are identical, in the sense that $\widetilde{\mathbf{v}}_T = \widetilde{\mathbf{w}}_T$. Moreover, when $T \geq \frac{\sqrt{8\left(4\log n \log T + \frac{2T}{q-1}\right)}}{\gamma}$, we have*

$$\frac{\min_{\mathbf{p} \in \Delta^n} \mathbf{p}^\top \mathbf{A} \widetilde{\mathbf{v}}_T}{\|\widetilde{\mathbf{v}}_T\|_q} \geq \gamma - \frac{\sum_{t=1}^{T} \|\mathbf{p}_t - \mathbf{p}_{t-1}\|_1^2}{\gamma^2(q-1)T^2} - \frac{32\log n \log T}{\gamma^2 T^2} \tag{8}$$

*and*

$$\left\| \frac{\widetilde{\mathbf{v}}_T}{\|\widetilde{\mathbf{v}}_T\|_q} - \mathbf{w}^*_{\|\cdot\|_q} \right\|_q \leq \frac{8\sqrt{2}}{\gamma^2\sqrt{(q-1)}} \sqrt{\frac{\sum_{t=1}^{T} \|\mathbf{p}_t - \mathbf{p}_{t-1}\|_1^2}{(q-1)T^2} + \frac{32\log n \log T}{T^2}}.$$

The above theorem shows that, for sufficiently large $T$, the margin maximization rate can be data-dependent. Note that $\sum_{t=1}^{T} \|\mathbf{p}_t - \mathbf{p}_{t-1}\|_1^2 \leq 2T$, so in the worst case, the bound reduces to the results in Theorem 3, but it can become significantly better when $\sum_{t=1}^{T} \|\mathbf{p}_t - \mathbf{p}_{t-1}\|_1^2$ is small. We expect that when $T$ is very large, $\mathbf{p}_T$ will change very slowly as we already know that the direction of $\widetilde{\mathbf{v}}_T$ will converge — however, turning this into a precise faster rate dependent on the original training data geometry, i.e. $\mathbf{A}$, is an intriguing open question.

**Algorithm 3** Steepest Descent [Recall $\ell_t(\mathbf{p}) = g(\mathbf{p}, \mathbf{w}_t)$, and $h_t(\mathbf{w}) = -g(\mathbf{p}_t, \mathbf{w})$]

| | |
|---|---|
| 1: **for** $t = 1, \ldots, T$ **do** 
 2: $\quad \mathbf{s}_{t-1} = \underset{\|\mathbf{s}\| \leq 1}{\operatorname{argmin}} \mathbf{s}^\top \nabla L(\mathbf{v}_{t-1})$ 
 3: $\quad \mathbf{v}_t = \mathbf{v}_{t-1} + \eta_{t-1} \mathbf{s}_{t-1}$ 
 4: **end for** 
 5: **Output**: $\mathbf{v}_T$ | $\mathbf{w}$-player: $\mathbf{w}_t = \underset{\mathbf{w} \in \mathbb{R}^d}{\operatorname{argmin}} \langle \delta_{t-1} \alpha_{t-1} \nabla h_{t-1}(\mathbf{w}_{t-1}), \mathbf{w} \rangle$ 
 $\qquad \qquad \qquad + D_{\frac{1}{2}\|\cdot\|^2}(\mathbf{w}, \mathbf{w}_{t-1})$ 
 $\mathbf{p}$-player: $\mathbf{p}_t = \underset{\mathbf{p} \in \Delta^n}{\operatorname{argmin}} \sum_{i=1}^t \alpha_i \ell_i(\mathbf{p}) + D_{\mathrm{KL}}\left(\mathbf{p}, \frac{1}{n}\right)$ 
 **Output**: $\widetilde{\mathbf{w}}_T = \sum_{t=1}^T \alpha_t \mathbf{w}_t$. |

## 4.2 Steepest Descent

Next, we consider the steepest descent method under a given *general* norm $\|\cdot\|$. For a succinct description of this algorithm see (Boyd & Vandenberghe, 2004). For completeness, we have also described this algorithm in the left box of Algorithm 3. In each iteration $t$, the algorithm first identifies the steepest direction with respect to the norm $\|\cdot\|$ (Step 2). It then adjusts the decision towards this direction using a specific step size $\eta_t$ (Step 3). After $T$ iterations, the algorithm yields the final iteration $\mathbf{v}_T$. In the following, we show that an $O\left(\frac{\lambda + \log n}{\gamma^2 \lambda T}\right) \|\cdot\|$-margin maximization rate can be achieved when the squared-norm, i.e. $\frac{1}{2}\|\cdot\|^2$ is $\lambda$-strongly convex (e.g., $\frac{1}{2}\|\cdot\|_q^2$ is $(q-1)$-strongly convex wrt $\|\cdot\|_q$). The proof of this result provided in Appendix D. Moreover, we recover the slower $O\left(\frac{\log n + \log T}{\sqrt{T}}\right)$ of Nacson et al. (2019) rate as a special case for norms that are not necessarily strongly convex.

**Theorem 5.** *Suppose Assumption 1 holds wrt a general norm $\|\cdot\|$, and $\frac{1}{2}\|\cdot\|^2$ is $\lambda$-strongly convex wrt $\|\cdot\|$. Let $\eta_t = \frac{\alpha_t \|\nabla L(\mathbf{w}_t)\|}{L(\mathbf{w}_t)}$, and $\delta_{t-1} = \alpha_{t-1}$. Then the methods in the two boxes of Algorithm 3 are are equivalent, in the sense that $\mathbf{v}_T = \widetilde{\mathbf{w}}_T$. Moreover, let $\alpha_t = \frac{\lambda}{2}$. Then $C_T = \frac{\frac{\lambda}{4} + \log n}{T\lambda}$. Therefore, when $T \geq \frac{\lambda + 4\log n}{\lambda \gamma^2}$, we have*

$$\frac{\min_{\mathbf{p} \in \Delta^n} \mathbf{p}^\top \mathbf{A} \mathbf{v}_T}{\|\mathbf{v}_T\|} \geq \gamma - \frac{\lambda + 4\log n}{\gamma^2 T\lambda},$$

*and*

$$\left\| \frac{\mathbf{v}_T}{\|\mathbf{v}_T\|} - \mathbf{w}^*_{\|\cdot\|} \right\| \leq \frac{4\sqrt{2}}{\sqrt{\lambda}\gamma^2} \sqrt{\frac{\lambda + 4\log n}{T\lambda}}.$$

The first part of Theorem 5 elucidates the equivalent online dynamic of the steepest descent algorithm, which is also depicted in the right box of Algorithm 3. The $\mathbf{w}$-player employs the standard online mirror descent (OMD) algorithm (Hazan, 2016), while the $\mathbf{p}$-player utilizes FTRL$^+$, i.e., $\mathbf{p}_t$ is selected by minimizing the cumulative loss observed so far, coupled with a regularization term. The crux of our algorithm equivalence analysis lies in evaluating the output of the $\mathbf{w}$-player. For this, we initially prove that given $\delta_t = \frac{1}{\alpha_t}$, the OMD algorithm condenses to best-response (BR), that is, $\mathbf{w}_t = \operatorname{argmin}_{\mathbf{w} \in \mathbb{R}^d} \alpha_t h_{t-1}(\mathbf{w})$. We then prove that BR's output coincides with the steepest descent direction. The second part of Theorem 5 shows that the average regret of this online learning dynamic is $O\left(\frac{\lambda + \log n}{\gamma^2 \lambda T}\right)$, which leads to the corresponding fast margin maximization/small direction error. We note that the favorable average regret is made possible by allowing the two players to play against each other, rather than plugging in worst-case regret bounds.

## 4.3 Even Faster Rates with Accelerated Generic Methods

In the preceding subsections, we showed that, with suitable step sizes, steepest descent and average mirror descent can achieve an $O\left(\frac{\log n \log T}{T}\right)$ margin maximization rate. We now aim to derive even faster rates using two approaches, as illustrated in the top two boxes of Algorithm 4. The left box introduces a Nesterov-acceleration-based mirror descent (Nesterov, 1988; Tseng, 2008): In each iteration $t$, the algorithm initially performs an extra update to yield $\mathbf{v}_t$ (Step 2), then executes a mirror descent step with the gradient at $\mathbf{v}_t$ (Step 3), and finally calculates a moving average (Step 4). On the other hand, the right box depicts a momentum-based steepest descent algorithm: In each iteration,

**Algorithm 4** Accelerated Methods [Recall $\ell_t(\mathbf{p}) = g(\mathbf{p}, \mathbf{w}_t)$, and $h_t(\mathbf{w}) = -g(\mathbf{p}_t, \mathbf{w})$]

| |
|---|
| 1: **for** $t = 1, \ldots, T$ **do** |
| 2: $\quad \mathbf{v}_t = \beta_{t,1}\widetilde{\mathbf{v}}_{t-1} + \beta'_{t,1}\mathbf{z}_{t-1}$ |
| 3: $\quad \nabla\Phi(\mathbf{z}_t) = \nabla\Phi(\mathbf{z}_{t-1}) - \eta_t\nabla L(\mathbf{v}_t)$ |
| 4: $\quad \widetilde{\mathbf{v}}_t = \beta_{t,2}\widetilde{\mathbf{v}}_{t-1} + \beta'_{t,2}\mathbf{z}_t$ |
| 5: **end for** |
| 6: **Output:** $\widetilde{\mathbf{v}}_T$ |

| |
|---|
| 1: **for** $t = 1, \ldots, T$ **do** |
| 2: $\quad \mathbf{g}_t = \beta_{t,3}\mathbf{g}_{t-1} + \beta'_{t,3}\nabla L(\beta_{t,4}\mathbf{v}_{t-1} + \beta'_{t,4}\mathbf{s}_{t-1})$ |
| 3: $\quad \mathbf{s}_t = \arg\min_{\|\mathbf{s}\|\le 1}\mathbf{s}^\top\mathbf{g}_t$ |
| 4: $\quad \mathbf{v}_t = \mathbf{v}_{t-1} + \eta_t\mathbf{s}_t$ |
| 5: **end for** |
| 6: **Output:** $\mathbf{v}_T$ |

$\mathbf{p}$-player: $\mathbf{p}_t = \arg\min_{\mathbf{p}\in\Delta^n}\sum_{i=1}^{t-1}\alpha_i\ell_i(\mathbf{p}) + \alpha_t\ell_{t-1}(\mathbf{p}) + \frac{1}{c}D_{\mathrm{KL}}(\mathbf{p}, \mathbf{p}_0)$

$\mathbf{w}$-player: $\mathbf{w}_t = \arg\min_{\mathbf{w}\in\mathbb{R}^d}\sum_{i=1}^{t}\alpha_i h_i(\mathbf{w})$

**Output:** $\widetilde{\mathbf{w}}_T = \sum_{t=1}^{T}\alpha_t\mathbf{w}_t$

the method maintains a momentum term $\mathbf{g}_t$ with an additional gradient (Step 2), then identifies the steepest direction with respect to $\mathbf{g}_t$ (Step 3), and applies this direction to update the decision (Step 4). At first glance, these two algorithms appear markedly different. However, we show that with appropriately chosen parameters, they are actually equivalent, in the sense that they both correspond to the online dynamic in the bottom box of Algorithm 4. More specifically, we provide the following theoretical guarantee. The proof is given in Appendix E.

**Theorem 6.** *Suppose Assumption 1 holds wrt a general norm $\|\cdot\|$, and $\frac{1}{2}\|\cdot\|^2$ is $\lambda$-strongly convex wrt $\|\cdot\|$. For the left box, let $\beta_{t,1} = \frac{\lambda}{4}$, $\beta'_{t,1} = \frac{\lambda}{2(t-1)}$, $\beta_{t,2} = 1$, $\beta'_{t,2} = \frac{2}{t+1}$, and $\eta_t = \frac{t}{L(\mathbf{v}_t)}$. For the right box, let $\beta_{t,3} = \frac{t-1}{t+1}$, $\beta_{t,4} = \frac{\lambda}{4}$, $\beta'_{t,4} = \frac{\lambda t\|\mathbf{g}_{t-1}\|_*}{4}$, $\beta'_{t,3} = \frac{2}{(t+1)L(\beta_{t,4}\mathbf{v}_{t-1}+\beta'_{t,4}\mathbf{s}_{t-1})}$, and $\eta_t = t\|\mathbf{g}_t\|_*$. For the bottom box, let $c = \frac{\lambda}{4}$, $\alpha_t = t$. Then all three methods in Algorithm 4 are identical, in the sense that $\widetilde{\mathbf{v}}_T = \mathbf{v}_T = \widetilde{\mathbf{w}}_T$. Moreover, when $T \ge \frac{4\sqrt{2\log n}}{\sqrt{\lambda}\gamma}$, we have*

$$\frac{\min_{\mathbf{p}\in\Delta^n}\mathbf{p}^\top A\widetilde{\mathbf{w}}_T}{\|\widetilde{\mathbf{w}}_T\|} \ge \gamma - \frac{32\log n}{\gamma^2 T^2\lambda},$$

*and*

$$\left\|\frac{\mathbf{v}_T}{\|\mathbf{v}_T\|} - \mathbf{w}^*_{\|\cdot\|}\right\| \le \frac{32\sqrt{\log n}}{\gamma^2\lambda T}.$$

**Remark** Theorem 6 reveals that the two strategies implemented in Algorithm 4 yield an optimal $O(\log n/[\gamma^2 T^2])$ rate. It is worth noting that a similar online dynamic to the one detailed in the bottom box of Algorithm 4 was also considered by (Algorithm 5, Wang et al., 2022b). Nonetheless, there are some crucial distinctions: 1) Their work only demonstrated that this dynamic could achieve a positive margin, leaving open questions regarding whether the margin can be maximized (i.e., converge to $\gamma$), and if so, what the margin maximization rate would be; 2) They only presented the online dynamic, without its equivalent optimization form. Finally, we note that, Theorem 6 requires the norm to be strongly convex, which is satisfied for, e.g. the $q$-norm when $q \in (1, 2]$.

## 5 Conclusion and Future Work

This paper examines the implicit bias of generic optimization methods, delivering accelerated margin maximization and directional errors for average mirror descent and steepest descent. Our approach translates *generic optimization methods for ERM* into *online learning dynamics for a regularized bilinear game*, offering a simpler analysis and a fresh perspective on implicit bias. Despite the effectiveness of the game framework in handling generic methods and accelerated techniques, it presently holds some limitations: 1) the framework is currently operational only for exponential loss, making its extension to handle more general losses a vital area for future research; 2) identifying algorithmic equivalence is nuanced and non-trivial, and it is as yet unresolved whether this framework can elucidate other methods, such as the last-iterate of MD; 3) Finally, it remains to be seen whether more advanced online learning algorithms are beneficial under our framework, such as parameter-free online learning (Orabona & Pál, 2016; Cutkosky & Orabona, 2018) or adaptive online learning methods (van Erven & Koolen, 2016; Wang et al., 2019; Zhang et al., 2019; Wang et al., 2021a).

**Acknowledgments.** GW was supported by a ARC-ACO fellowship provided by Georgia Tech. VM was supported by the NSF (through CAREER award CCF-2239151 and award IIS-2212182), an Adobe Data Science Research Award, an Amazon Research Award and a Google Research Colabs Award. JA was supported by the AI4OPT Institute, as part of NSF Award 2112533, and NSF through Award IIS-1910077.

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

# A  Additional Related Work

As discussed in the introduction, the implicit bias of GD-based methods on linear spreadable data with exponentially-tailed loss has been extensively studied (Soudry et al., 2018; Ji & Telgarsky, 2018; Nacson et al., 2019; Ji & Telgarsky, 2021; Ji et al., 2021; Wang et al., 2022b). For $\ell_1$-norm, Telgarsky (2013) provides an $O(\frac{1}{\sqrt{T}})$ margin maximization rate, and an $O(\frac{1}{T})$ convergence rate to a sub-optimal margin. The idea of using the regularized game framework for analyzing the implicit bias of Nesterov-accelerated GD is firstly considered in Wang et al. (2022b). This idea is motivated by the smooth perceptron analysis of Soheili & Pena (2012), and the the Fenchel game framework by Wang et al. (2021b). The idea also inspired by the primal-dual analysis of Ji & Telgarsky (2021), who discovered the relationship between the normalized gradient descent for the exponential loss and a smooth update of a distribution over data. Apart from the setting above, the implicit bias of GD-based optimization methods have also been studied in other cases, such as minimizing more general loss functions (Ji et al., 2020; Ji & Telgarsky, 2021; Lai & Muthukumar, 2023), and deep neural networks (Gunasekar et al., 2018b; Ji & Telgarsky, 2020a,b; Lyu & Li, 2020; Vardi, 2022).

Compared with GD-based methods, the implicit bias of generic optimization methods for linear classification is less understood. Apart from the work summarized in the introduction (Gunasekar et al., 2018a; Nacson et al., 2019; Li et al., 2021; Sun et al., 2022), the implicit bias of mirror descent is also studied in regression problems (Gunasekar et al., 2018a; Azizan & Hassibi, 2019). We note that, as indicated in Gunasekar et al. (2018a) and Sun et al. (2022), the analysis of implicit bias for classification and regression are "fundamentally different". Apart from steepest and mirror descent, the implicit bias of other generic methods, such as Adagrad and Adam are also studied (Gunasekar et al., 2018a; Wang et al., 2022a)

**Discussion on large enough $T$**   Our Theorem 1 indicates that to ensure the margin can be maximized, the number of iterations $T$ has to be large enough such that $C_T \leq \frac{\gamma^2}{4}$. We note that, similar conditions are also required (explicitly or implicitly) in other work for analyzing the margin maximization rates for generic optimization methods. For example, in the proof of Theorem 8 of Nacson et al. (2019), in order to combine the upper bound in (53) and lower bound in (54), one need to make sure the LHS of (53), i.e., the margin, is non-negative, which essentially hinge on the condition that $\sqrt{T} = \Omega\left(\frac{\log n \log T}{\gamma^2}\right)$, similar to what is required in our Corollary 1. In Sun et al. (2022), page 21, to make sure the term $1 + \frac{\log n}{m_t}$ be a constant, $T$ has to be large enough such that $m_t$, the margin, goes to $\infty$. In Li et al. (2021), the requirements for a large enough $T$ is stated in the main theorem (e.g., Theorem 4.2).

**Comparison with Wang et al. (2022b)**   As discussed in the introduction, our work is motivated by Wang et al. (2022b). Compared with Wang et al. (2022b), we note that: 1) Wang et al. (2022b) draws the connection between Nesterov-accelerated GD for ERM and solving the bilinear game through an online dynamic. However, it was unclear whether this kind of analysis suits other gradient-descent-based methods, and generic optimization methods such as mirror descent/steepest descent was not addressed at all. We observe in this work that the non-linearity of the mirror map in generic optimization methods, such as mirror descent and steepest descent, makes the analysis particularly challenging. In this paper, we reveal that the game framework can in fact encompass implicit bias analysis for a range of generic optimization methods, and offer a more streamlined and unified analysis. 2) Wang et al. (2022b) also proposed an accelerated p-norm perceptron problem. However, they only demonstrated that the algorithm could achieve a non-negative margin, leaving open questions regarding whether the margin can be maximized, and if so, what the margin maximization rate would be; 2) They only presented the online dynamic, without its equivalent optimization form under ERM.

# B  Proof of Theorem 1

Define $m(\mathbf{w}) = \min_{\mathbf{p} \in \Delta^n} g(\mathbf{p}, \mathbf{w})$, $\overline{\mathbf{w}}_T = \frac{1}{\sum_{t=1}^T \alpha_t} \sum_{t=1}^T \alpha_t \mathbf{w}_t = \frac{1}{\sum_{t=1}^T \alpha_t} \widetilde{\mathbf{w}}_T$, and we introduce following lemma, which shows that using online learning for solving the game defined in (2) maximizes $m(\mathbf{w})$.

**Lemma 1** (Abernethy et al. (2018)). *Consider solving the game defined in* (2) *with the online learning dynamic defiend in Protocol 1. We have that* $\forall \mathbf{w} \in \mathbb{R}^d$,

$$m(\mathbf{w}) - m(\overline{\mathbf{w}}_T) \leq \frac{\mathrm{Reg}_T^{\mathbf{P}} + \mathrm{Reg}_T^{\mathbf{w}}}{\sum_{t=1}^{T} \alpha_t}.$$

Based on Lemma 1 and the definition of $m(\cdot)$, we have

$$
\begin{aligned}
m\left(\overline{\mathbf{w}}_T\right) = m\left(\frac{\widetilde{\mathbf{w}}_T}{\sum_{t=1}^{T} \alpha_t}\right) &= \min_{\mathbf{p} \in \Delta^n} \mathbf{p}^\top \mathbf{A} \frac{\widetilde{\mathbf{w}}_T}{\sum_{t=1}^{T} \alpha_t} - \frac{1}{2}\left\|\frac{\widetilde{\mathbf{w}}_T}{\sum_{t=1}^{T} \alpha_t}\right\|^2 \\
&\geq m\left(\left\|\frac{\widetilde{\mathbf{w}}_T}{\sum_{t=1}^{T} \alpha_t}\right\| \mathbf{w}^*_{\|\cdot\|}\right) - \frac{\mathrm{Reg}_T^{\mathbf{P}} + \mathrm{Reg}_T^{\mathbf{w}}}{\sum_{t=1}^{T} \alpha_t} \\
&= \gamma\left\|\frac{\widetilde{\mathbf{w}}_T}{\sum_{t=1}^{T} \alpha_t}\right\| - \frac{1}{2}\left\|\frac{\widetilde{\mathbf{w}}_T}{\sum_{t=1}^{T} \alpha_t}\right\|^2 - \frac{\mathrm{Reg}_T^{\mathbf{P}} + \mathrm{Reg}_T^{\mathbf{w}}}{\sum_{t=1}^{T} \alpha_t},
\end{aligned}
\tag{9}
$$

which implies that

$$\frac{\min_{\mathbf{p} \in \Delta^n} \mathbf{p}^\top \mathbf{A} \widetilde{\mathbf{w}}_T}{\|\widetilde{\mathbf{w}}_T\|} \geq \gamma - \frac{\mathrm{Reg}_T^{\mathbf{P}} + \mathrm{Reg}_T^{\mathbf{w}}}{\|\widetilde{\mathbf{w}}_T\|}. \tag{10}$$

The above proof follows the main idea in Wang et al. (2022b). Next, we turn to lower bound $\|\widetilde{\mathbf{w}}_T\|$. Note that since $\mathbf{w}_T$ (and also $\widetilde{\mathbf{w}}_T$) does not have a simple explicit form, the the technique for lower bounding the norm in (Wang et al., 2022b) fails and we need to find a new approach. Let $(\mathbf{x}, y) \in \{(\mathbf{x}^{(i)}, y^{(i)})\}_{i=1}^n$ be a data point. We have

$$\|\widetilde{\mathbf{w}}_T\| \geq \|y\mathbf{x}\|_* \|\widetilde{\mathbf{w}}_T\| \geq y\mathbf{x}^\top \widetilde{\mathbf{w}}_T \geq \min_{\mathbf{p} \in \Delta^n} \mathbf{p}^\top \mathbf{A} \widetilde{\mathbf{w}}_T, \tag{11}$$

where the first inequality is due to assumption that $\|\mathbf{x}\|_* \leq 1$, the second inequality is derived from the Cauchy-Schwarz inequality. To proceed, we need a lower bound on the unnormalized margin of $\widetilde{\mathbf{w}}_T$. We have

$$
\begin{aligned}
m\left(\overline{\mathbf{w}}_T\right) = m\left(\frac{\widetilde{\mathbf{w}}_T}{\sum_{t=1}^{T} \alpha_t}\right) &= \min_{\mathbf{p} \in \Delta^n} \mathbf{p}^\top \mathbf{A} \frac{\widetilde{\mathbf{w}}_T}{\sum_{t=1}^{T} \alpha_t} - \frac{1}{2}\left\|\frac{\widetilde{\mathbf{w}}_T}{\sum_{t=1}^{T} \alpha_t}\right\|^2 \\
&\geq m\left(\gamma \mathbf{w}^*_{\|\cdot\|}\right) - \frac{\mathrm{Reg}_T^{\mathbf{P}} + \mathrm{Reg}_T^{\mathbf{w}}}{\sum_{t=1}^{T} \alpha_t} \\
&= \min_{\mathbf{p} \in \Delta^n} \mathbf{p}^\top \mathbf{A} \mathbf{w}^*_{\|\cdot\|} - \frac{1}{2}\|\gamma \mathbf{w}^*_{\|\cdot\|}\|^2 - \frac{\mathrm{Reg}_T^{\mathbf{P}} + \mathrm{Reg}_T^{\mathbf{w}}}{\sum_{t=1}^{T} \alpha_t} \\
&= \frac{\gamma^2}{2} - \frac{\mathrm{Reg}_T^{\mathbf{P}} + \mathrm{Reg}_T^{\mathbf{w}}}{\sum_{t=1}^{T} \alpha_t},
\end{aligned}
\tag{12}
$$

where for the first inequality we apply Lemma 1 and compare $m(\widetilde{\mathbf{w}}_T)$ with that of $\gamma \mathbf{w}^*_{\|\cdot\|}$, and the last equality is derived based on Assumption 1 (the margin of $\mathbf{w}^*_{\|\cdot\|}$ is $\gamma$, and $\|\mathbf{w}^*_{\|\cdot\|}\| = 1$). . (12) suggests that

$$\min_{\mathbf{p} \in \Delta^n} \mathbf{p}^\top \mathbf{A} \widetilde{\mathbf{w}}_T \geq \underbrace{\frac{1}{2}\frac{\|\widetilde{\mathbf{w}}_T\|^2}{\sum_{t=1}^{T} \alpha_t}}_{\geq 0} + \frac{\gamma^2}{2}\sum_{t=1}^{T} \alpha_t - (\mathrm{Reg}_T^{\mathbf{P}} + \mathrm{Reg}_T^{\mathbf{w}}) \geq \frac{\gamma^2}{2}\sum_{t=1}^{T} \alpha_t - (\mathrm{Reg}_T^{\mathbf{P}} + \mathrm{Reg}_T^{\mathbf{w}}).$$

$$\tag{13}$$

Note that, to plug in the lower bound of $\widetilde{\mathbf{w}}_T$, we need to ensure the RHS of (13) is positive. Notice that when $\frac{\gamma^2}{2}\sum_{t=1}^{T} \alpha_t \geq 2(\mathrm{Reg}_T^{\mathbf{P}} + \mathrm{Reg}_T^{\mathbf{w}})$,

$$\|\widetilde{\mathbf{w}}_T\| \geq \frac{\gamma^2}{2}\sum_{t=1}^{T} \alpha_t - (\mathrm{Reg}_T^{\mathbf{P}} + \mathrm{Reg}_T^{\mathbf{w}}) \geq \frac{\gamma^2}{4}\sum_{t=1}^{T} \alpha_t + \left[\frac{\gamma^2}{4}\sum_{t=1}^{T} \alpha_t - (\mathrm{Reg}_T^{\mathbf{P}} + \mathrm{Reg}_T^{\mathbf{w}})\right] \geq \frac{\gamma^2}{4}\sum_{t=1}^{T} \alpha_t.$$

Combining (10), (11), and (13), we have

$$\frac{\min_{\mathbf{p}\in\Delta^n} \mathbf{p}^\top \mathbf{A}\widetilde{\mathbf{w}}_T}{\|\widetilde{\mathbf{w}}_T\|} \geq \gamma - \frac{4\left(\mathrm{Reg}_T^{\mathbf{P}} + \mathrm{Reg}_T^{\mathbf{w}}\right)}{\gamma^2 \sum_{t=1}^T \alpha_t} = \frac{4C_T}{\gamma^2}. \tag{14}$$

Note that to apply (13), we need $\frac{\gamma^2}{2}\sum_{t=1}^T \alpha_t - 2\left(\mathrm{Reg}_T^{\mathbf{P}} + \mathrm{Reg}_T^{\mathbf{w}}\right) \geq 0$.

Finally, we focus on the distance between $\frac{\widetilde{\mathbf{w}}_T}{\|\widetilde{\mathbf{w}}_T\|}$ and $\mathbf{w}_{\|\cdot\|}^*$ for the case where $\Phi(\mathbf{w})$ is strongly convex wrt $\|\cdot\|$. This part of the proof is motivated by Theorem 4 of Ramdas & Pena (2016), who show that a variant of the perceptron algorithm can converge to the $\ell_2$-maximum margin classifier in an $O(1/\sqrt{t})$ convergence rate. We have

$$\begin{aligned}
\left\|\frac{\widetilde{\mathbf{w}}_T}{\|\widetilde{\mathbf{w}}_T\|} - \mathbf{w}_{\|\cdot\|}^*\right\| = \left\|\frac{\overline{\mathbf{w}}_T}{\|\overline{\mathbf{w}}_T\|} - \mathbf{w}_{\|\cdot\|}^*\right\| &= \frac{\left\|\overline{\mathbf{w}}_T - \|\overline{\mathbf{w}}_T\|\mathbf{w}_{\|\cdot\|}^*\right\|}{\|\overline{\mathbf{w}}_T\|} \\
&= \frac{\left\|\overline{\mathbf{w}}_T - \gamma\mathbf{w}_{\|\cdot\|}^* + \gamma\mathbf{w}_{\|\cdot\|}^* - \|\overline{\mathbf{w}}_T\|\mathbf{w}_{\|\cdot\|}^*\right\|}{\|\mathbf{w}\|} \\
&\leq \frac{\left\|\overline{\mathbf{w}}_T - \gamma\mathbf{w}_{\|\cdot\|}^*\right\| + \left|\gamma - \|\overline{\mathbf{w}}_T\|\right|}{\|\overline{\mathbf{w}}_T\|} \\
&= \frac{\left\|\overline{\mathbf{w}}_T - \gamma\mathbf{w}_{\|\cdot\|}^*\right\| + \left|\|\gamma\mathbf{w}_{\|\cdot\|}^*\| - \|\overline{\mathbf{w}}_T\|\right|}{\|\overline{\mathbf{w}}_T\|} \\
&\leq \frac{2\left\|\overline{\mathbf{w}}_T - \gamma\mathbf{w}_{\|\cdot\|}^*\right\|}{\|\overline{\mathbf{w}}_T\|}
\end{aligned} \tag{15}$$

where the first inequality is based on the Minkowski inequality and the fact that $\|\mathbf{w}_{\|\cdot\|}^*\| = 1$. Next, note that $m(\mathbf{w})$ is $\lambda$-strongly concave with respect to $\|\cdot\|$, and $\gamma\mathbf{w}_{\|\cdot\|}^*$ maximize $m(\mathbf{w})$. This is because it is easy to see that the optimal solution of $m(\mathbf{w})$ always lies in the direction of $\mathbf{w}_{\|\cdot\|}^*$, and we only need to decide the norm. Let $c > 0$ be some constant, we have $m\left(c\mathbf{w}_{\|\cdot\|}^*\right) = c\gamma - \frac{1}{2}c^2$. The function is maximized when $c = \gamma$, which implies that the optimal solution is $\gamma\mathbf{w}_{\|\cdot\|}^*$. Combining these facts with Lemma 1, we have

$$\frac{\lambda}{2}\|\overline{\mathbf{w}}_T - \gamma\mathbf{w}_{\|\cdot\|}^*\|^2 \leq m(\gamma\mathbf{w}_{\|\cdot\|}^*) - m(\overline{\mathbf{w}}_T) \leq \frac{\mathrm{Reg}_T^{\mathbf{P}} + \mathrm{Reg}_T^{\mathbf{w}}}{\sum_{t=1}^T \alpha_t}. \tag{16}$$

Finally, combining the lower bound of $\overline{\mathbf{w}}_T$ proved in (11), we have when $\frac{\gamma^2}{2}\sum_{t=1}^T \alpha_t - 2\left(\mathrm{Reg}_T^{\mathbf{P}} + \mathrm{Reg}_T^{\mathbf{w}}\right) \geq 0$,

$$\|\overline{\mathbf{w}}_T\| = \frac{1}{\sum_{t=1}^T \alpha_t}\|\widetilde{\mathbf{w}}_T\| \geq \frac{1}{\sum_{t=1}^T \alpha_t}\left[\frac{\gamma^2}{4}\sum_{t=1}^T \alpha_t\right] = \frac{\gamma^2}{4},$$

so, combining the equation above with (15) and (16), we get

$$\left\|\frac{\widetilde{\mathbf{w}}_T}{\|\widetilde{\mathbf{w}}_T\|} - \mathbf{w}_{\|\cdot\|}^*\right\| \leq \frac{8\sqrt{2(\mathrm{Reg}_T^{\mathbf{P}} + \mathrm{Reg}_T^{\mathbf{w}})}}{\gamma^2\sqrt{\lambda\sum_{t=1}^T \alpha_t}}. \tag{17}$$

## C  Omitted Proof in Section 4.1

In this section, we provide the omitted proof of Section 4.1. Here, we present a more general algorithm framework (given in Algorithm 5) which allows different step sizes. In the following, we first state a general theorem for this algorithm, and Theorems 2 and 3 is then given as Corollaries 2 and 3.

**Algorithm 5** Mirror Descent (General Version)

| | |
|---|---|
| 1: **for** $t = 1, \ldots, T$ **do** | **p**-player: $\mathbf{p}_t = \underset{\mathbf{p} \in \Delta^n}{\operatorname{argmin}} \, \alpha_t \ell_{t-1}(\mathbf{p}) + \beta_t D_{\mathrm{KL}}\left(\mathbf{p}, \frac{1}{n}\right)$ |
| 2: $\quad \nabla \Phi(\mathbf{v}_t) = \nabla \Phi(\mathbf{v}_{t-1}) - \eta_t \nabla L(\mathbf{v}_{t-1})$ | **w**-player: $\mathbf{w}_t = \underset{\mathbf{w} \in \mathbb{R}^d}{\operatorname{argmin}} \sum_{j=1}^t \alpha_j h_j(\mathbf{w})$ |
| 3: **end for** | |
| 4: **Output:** $\widetilde{\mathbf{v}}_T = \sum_{t=1}^T \frac{\alpha_t}{\sum_{i=1}^t \alpha_i} \mathbf{v}_t$ | **Output:** $\widetilde{\mathbf{w}}_T = \sum_{t=1}^T \alpha_t \mathbf{w}_t$ |

**Theorem 7.** *Suppose Assumption 1 holds wrt $\|\cdot\|_q$-norm for $q \in (1, 2]$. For the left box of Algorithm 5, let $\eta_t = \frac{\alpha_t}{L(\mathbf{v}_{t-1})}$. For the right box, let $\beta_t$ be $\frac{\alpha_t}{\sum_{i=1}^{t-1} \alpha_i}$ for $t > 1$, $\beta_1 = \alpha_1$. Then the methods in the two boxes of Algorithm 5 are identical, in the sense that $\widetilde{\mathbf{v}}_T = \widetilde{\mathbf{w}}_T$, and $\mathbf{v}_t = \mathbf{w}_t \cdot \sum_{i=1}^t \alpha_i$. Moreover, the regret upper bound*

$$\mathrm{Reg}_T^{\mathbf{p}} = 2 \sum_{t=2}^T \frac{\alpha_t^2}{\sum_{j=1}^{t-1} \alpha_j (q-1)} + \sum_{t=2}^T \frac{\sum_{j=1}^{t-1} \alpha_j (q-1)}{\cdot 2} \|\mathbf{w}_t - \mathbf{w}_{t-1}\|_q^2 + 2 \log n \sum_{t=1}^T \beta_t + \alpha_1,$$

$$\mathrm{Reg}_T^{\mathbf{w}} = -\sum_{t=2}^T \left( \frac{(q-1) \sum_{s=1}^{t-1} \alpha_s}{2} \right) \|\mathbf{w}_t - \mathbf{w}_{t-1}\|_q^2.$$

*Thus, $\widetilde{\mathbf{v}}_T$ achieves a positive margin (no smaller than $\gamma^2/4$) for sufficiently large $T$ such that*

$$\frac{\gamma^2}{4} \sum_{t=1}^T \alpha_t \geq \left( 2 \sum_{t=2}^T \frac{\alpha_t^2}{\sum_{j=1}^{t-1} \alpha_j (q-1)} + 2 \log n \sum_{t=2}^T \frac{\alpha_t}{\sum_{i=1}^{t-1} \alpha_i} \right) + \alpha_1 (1 + 2 \log n). \qquad (18)$$

*After (18) is satisfied, the margin of $\widetilde{\mathbf{v}}_T$ is lower bounded by*

$$\frac{\underset{\mathbf{p} \in \Delta^n}{\min} \, \mathbf{p}^\top A \widetilde{\mathbf{v}}_T}{\|\widetilde{\mathbf{v}}_T\|_q} \geq \gamma - \frac{4 \left[ \left( 2 \sum_{t=2}^T \frac{\alpha_t^2}{\sum_{j=1}^{t-1} \alpha_j (q-1)} + 2 \log n \sum_{t=2}^T \frac{\alpha_t}{\sum_{i=1}^{t-1} \alpha_i} \right) + \alpha_1 (1 + 2 \log n) \right]}{\gamma^2 \sum_{t=1}^T \alpha_t},$$

*and the directional error is*

$$\left\| \frac{\widetilde{\mathbf{w}}_T}{\|\widetilde{\mathbf{w}}_T\|_q} - \mathbf{w}_{\|\cdot\|}^* \right\|_q$$

$$\leq \frac{8}{\gamma^2 \sqrt{q-1}} \sqrt{\frac{2 \left[ \left( 2 \sum_{t=2}^T \frac{\alpha_t^2}{\sum_{j=1}^{t-1} \alpha_j (q-1)} + 2 \log n \sum_{t=2}^T \frac{\alpha_t}{\sum_{i=1}^{t-1} \alpha_i} \right) + \alpha_1 (1 + 2 \log n) \right]}{(q-1) \sum_{t=1}^T \alpha_t}}.$$

Next, we show that different step sizes (decided by $\alpha_t$) lead to different implicit bias convergence rates. Firstly, we consider setting the step size as $\eta_t = \frac{1}{\sqrt{t} L(\mathbf{v}_{t-1})}$, which leads to a slow $\widetilde{O}(\frac{1}{\gamma^2 \sqrt{T}(q-1)})$ bound.

**Corollary 1.** *Let $\alpha_t = \frac{1}{\sqrt{t}}$. Then the margin is lower bounded by*

$$\frac{\underset{\mathbf{p} \in \Delta^n}{\min} \, \mathbf{p}^\top \mathbf{A} \widetilde{\mathbf{v}}_T}{\|\widetilde{\mathbf{v}}_T\|_q} \geq \gamma - \frac{4 \left( \left( \frac{2}{q-1} + \log n \right) \log T + 4 \log n + \frac{4}{q-1} \right)}{\gamma^2 \sqrt{T}} = \gamma - \widetilde{O} \left( \frac{1}{\gamma^2 \sqrt{T}(q-1)} \right),$$

*and*

$$\left\| \frac{\widetilde{\mathbf{w}}_T}{\|\widetilde{\mathbf{w}}_T\|_q} - \mathbf{w}_{\|\cdot\|}^* \right\|_q \leq \frac{8\sqrt{2}}{\gamma^2 \sqrt{q-1}} \sqrt{\frac{\left( \frac{2}{q-1} + \log n \right) \log T + 4 \log n + \frac{4}{q-1}}{\sqrt{T}}} = \widetilde{O} \left( \frac{1}{(q-1)\gamma^2 T^{1/4}} \right),$$

*when $T$ is sufficiently large such that $\sqrt{T} \geq \frac{4 \left( \left( \frac{2}{q-1} + \log n \right) \log T + 4 \log n + \frac{4}{q-1} \right)}{\gamma^2} = \widetilde{\Omega} \left( \frac{1}{(q-1)\gamma^2} \right)$.*

Next, we show that a faster rate can be obtained with a constant $\alpha_t$, which implies Theorem 2.

**Corollary 2** (Theorem 2). *Let $\alpha_t = 1$. Then the margin is lower bounded by*

$$\frac{\min_{\mathbf{p}\in\Delta^n} \mathbf{p}^\top \mathbf{A}\widetilde{\mathbf{v}}_T}{\|\widetilde{\mathbf{v}}_T\|_q} \geq \gamma - \frac{4\left(\frac{2}{q-1} + 2\log n\right)(\log T + 2)}{\gamma^2 T} = \gamma - \widetilde{O}\left(\frac{1}{(q-1)\gamma^2 T}\right),$$

*and*

$$\left\|\frac{\widetilde{\mathbf{w}}_T}{\|\widetilde{\mathbf{w}}_T\|_q} - \mathbf{w}^*_{\|\cdot\|}\right\|_q \leq \frac{8\sqrt{2}}{\gamma^2\sqrt{(q-1)}}\sqrt{\frac{\left(\frac{2}{q-1} + 2\log n\right)(\log T + 2)}{T}} = \widetilde{O}\left(\frac{1}{\gamma^2(q-1)\sqrt{T}}\right).$$

*when $T$ is sufficiently large such that $T \geq \frac{4\left(\frac{2}{q-1}+2\log n\right)(\log T+2)}{\gamma^2}$.*

**Remark** Note that, by setting $\alpha_t = 1$, according to Theorem 7, we get $\eta_t = \frac{1}{L(\mathbf{v}_{t-1})}$, $\beta_1 = 1$, and $\beta_t = \frac{1}{\sum_{j=1}^{t-1}\alpha_j} = \frac{1}{t-1}$, which recovers the parameter configuration in Theorem 2.

Corollary 2 shows that, with a step size of order $O\left(\frac{1}{L(\mathbf{v}_{t-1})}\right)$, Algorithm 1 enjoys a faster $\widetilde{O}(\frac{1}{(q-1)\gamma^2 t})$ margin maximization rate when $t = \Omega\left(\frac{\log n \log t}{(q-1)\gamma^2}\right)$. Next, we also considered using a even larger step size, $\eta_t = \frac{t}{L(\mathbf{v}_{t-1})}$, and recover Theorem 3.

**Corollary 3** (Theorem 3). *Let $\alpha_t = t$. Then $\widetilde{\mathbf{v}}_T = \sum_{t=1}^T \frac{2}{t+1}\mathbf{v}_t$. When $T \geq \sqrt{\frac{8\left[\frac{4T}{q-1}+4\log n\log T+1+2\log n\right]}{\gamma^2}}$, we have we have*

$$\frac{\min_{\mathbf{p}\in\Delta^n}\mathbf{p}^\top\mathbf{A}\widetilde{\mathbf{v}}_T}{\|\widetilde{\mathbf{v}}_T\|_q} \geq \gamma - \frac{32}{\gamma^2 T(q-1)} - \frac{8(4\log n\log T + 1 + 2\log n)}{\gamma^2 T^2}, \tag{19}$$

*and*

$$\left\|\frac{\widetilde{\mathbf{v}}_T}{\|\widetilde{\mathbf{v}}_T\|_q} - \mathbf{w}^*_{\|\cdot\|_q}\right\|_q \leq \frac{8\sqrt{2}}{\gamma^2\sqrt{q-1}}\sqrt{\frac{8}{(q-1)T} + \frac{4\log n\log T + 2\log n + 1}{T^2}}.$$

Note that, when $\alpha_t = t$, we have $\eta_t = \frac{t}{L(\mathbf{v}_{t-1})}$, which recovers Theorem 3. Finally, we note that, if $T$ is fixed and known before, we can set $\alpha_t$ as T and get rid of the $\log T$ term.

## C.1 Proof of Theorem 7

We first focus on the algorithm equivalence, and start from the online learning framework. For $\mathbf{w}$-player, we have

$$\begin{aligned}
\mathbf{w}_t &= \operatorname*{argmin}_{\mathbf{w}\in\mathbb{R}^d} \sum_{j=1}^t \alpha_j h_j(\mathbf{w}) \\
&= \operatorname*{argmin}_{\mathbf{w}\in\mathbb{R}^d} \sum_{j=1}^t -\alpha_j \mathbf{p}_j^\top \mathbf{A}\mathbf{w} + \frac{\sum_{j=1}^t \alpha_j}{2}\|\mathbf{w}\|_q^2 \\
&= [\nabla\Phi]^{-1}\left(\frac{1}{\sum_{j=1}^t \alpha_j}\sum_{j=1}^t \alpha_j \mathbf{A}^\top \mathbf{p}_j\right),
\end{aligned} \tag{20}$$

which implies that

$$\nabla\Phi(\mathbf{w}_t) = \frac{1}{\sum_{j=1}^t \alpha_j}\sum_{j=1}^t \alpha_j \mathbf{A}^\top \mathbf{p}_j = \frac{\sum_{j=1}^{t-1}\alpha_j}{\sum_{j=1}^t \alpha_j}\nabla\Phi(\mathbf{w}_{t-1}) + \frac{\alpha_t}{\sum_{j=1}^t \alpha_j}\mathbf{A}^\top \mathbf{p}_t.$$

To proceed, note that

$$[\nabla\Phi(\mathbf{w})]_i = \frac{\operatorname{sign}(w_i)|w_i|^{q-1}}{\|\mathbf{w}\|^{q-2}}.$$

Thus, $\forall c > 0$, $c\nabla\Phi(\mathbf{w}) = \nabla\Phi(c\mathbf{w})$. Then, multiplying $c = \sum_{j=1}^{t} \alpha_j$ on both sides, we get

$$\nabla\Phi\left(\mathbf{w}_t \sum_{j=1}^{t} \alpha_j\right) = \nabla\Phi\left(\mathbf{w}_{t-1} \sum_{j=1}^{t-1} \alpha_j\right) + \alpha_t \mathbf{A}^\top \mathbf{p}_t.$$

On the other hand, for the $\mathbf{p}$-player, we have

$$\mathbf{p}_t = \underset{\mathbf{p}\in\Delta^n}{\operatorname{argmin}} \, \alpha_t \ell_{t-1}(\mathbf{p}) + \beta_t D_{\text{KL}}\left(\mathbf{p}, \frac{\mathbf{1}}{n}\right) = \underset{\mathbf{p}\in\Delta^n}{\operatorname{argmin}} \, \frac{\alpha_t}{\beta_t} \mathbf{p}^\top \mathbf{A} \mathbf{w}_{t-1} + \sum_{i=1}^{n} p_i \log \frac{p_i}{\frac{1}{n}}.$$

Based on a standard argument on the relationship between OMD with the negative entropy regularizer on the simplex (see, e.g., Section 6.6 of Orabona, 2019), it is easy to verify that $\forall i \in [n], t \in [T]$,

$$p_{t,i} = \frac{\exp(-\frac{\alpha_t}{\beta_t} y^{(i)} \mathbf{x}^{(i)\top} \mathbf{w}_{t-1})}{\sum_{j=1}^{n} \exp(-\frac{\alpha_t}{\beta_t} y^{(j)} \mathbf{x}^{(j)\top} \mathbf{w}_{t-1})},$$

where $p_{t,i}$ is the $i$-th element of $\mathbf{p}_t$. Moreover, based on the definition of $L(\mathbf{w})$, for any $\mathbf{w} \in \mathbb{R}^d$,

$$\frac{\nabla L(\mathbf{w})}{L(\mathbf{w})} = -A^\top \left[\cdots, \frac{\exp(-y^{(i)} \mathbf{x}^{(i)\top} \mathbf{w})}{\sum_{j=1}^{n} \exp(-y^{(j)} \mathbf{x}^{(j)\top} \mathbf{w})}, \cdots\right]^\top,$$

which implies that

$$\mathbf{A}^\top \mathbf{p}_t = -\frac{\nabla L\left(\frac{\alpha_t}{\beta_t} \mathbf{w}_{t-1}\right)}{L\left(\frac{\alpha_t}{\beta_t} \mathbf{w}_{t-1}\right)}.$$

Combining the above equations and the definition of $\beta_t = \frac{\alpha_t}{\sum_{i=1}^{t-1} \alpha_i}$, we get

$$\nabla\Phi\left(\mathbf{w}_t \sum_{j=1}^{t} \alpha_j\right) = \nabla\Phi\left(\mathbf{w}_{t-1} \sum_{j=1}^{t-1} \alpha_j\right) - \alpha_t \frac{\nabla L\left(\mathbf{w}_{t-1} \sum_{j=1}^{t-1} \alpha_j\right)}{L\left(\mathbf{w}_{t-1} \sum_{j=1}^{t-1} \alpha_j\right)}.$$

Substituting $\mathbf{v}_t = \mathbf{w}_t \cdot \sum_{j=1}^{t} \alpha_j$, we get

$$\nabla\Phi(\mathbf{v}_t) = \nabla\Phi(\mathbf{v}_{t-1}) - \alpha_t \frac{\nabla L(\mathbf{v}_{t-1})}{L(\mathbf{v}_{t-1})},$$

and $\widetilde{\mathbf{w}}_T = \sum_{t=1}^{T} \alpha_t \mathbf{w}_t = \sum_{t=1}^{T} \frac{\alpha_t}{\sum_{j=1}^{t} \alpha_j} \mathbf{v}_t$. The proof is finished by replacing $\frac{\alpha_t}{L(\mathbf{v}_{t-1})}$ with $\eta_t$.

Next, we focus on bounding the regret of the two players. For the $\mathbf{p}$-player, let $\mathbf{p}^{*,\ell} = \min_{\mathbf{p}\in\Delta^n} \sum_{t=1}^{T} \alpha_t \mathbf{p}^\top \mathbf{A}\mathbf{w}_t$ be the best decision in hindsight for the online leanring problem. We have

$$
\begin{aligned}
\text{Reg}_T^{\mathbf{P}} &= \sum_{t=1}^{T} \alpha_t \mathbf{p}_t^\top \mathbf{A}\mathbf{w}_t - \sum_{t=1}^{T} \alpha_t \mathbf{p}^{*,\ell,\top} \mathbf{A}\mathbf{w}_t \\
&= \sum_{t=1}^{T} \left( \alpha_t \mathbf{p}_t^\top \mathbf{A}\mathbf{w}_{t-1} + \beta_t D_{\text{KL}}\left(\mathbf{p}_t, \frac{\mathbf{1}}{n}\right) \right) - \sum_{t=1}^{T} \alpha_t \mathbf{p}^{*,\ell,\top} \mathbf{A}\mathbf{w}_t + \sum_{t=1}^{T} \alpha_t \mathbf{p}_t^\top \mathbf{A}(\mathbf{w}_t - \mathbf{w}_{t-1}) \\
&\quad - \sum_{t=1}^{T} \beta_t D_E\left(\mathbf{p}_t, \frac{\mathbf{1}}{n}\right) \\
&\overset{(a)}{\leq} \sum_{t=1}^{T} \left( \alpha_t \mathbf{p}^{*,\ell,\top} \mathbf{A}\mathbf{w}_{t-1} + \beta_t D_{\text{KL}}\left(\mathbf{p}^{*,\ell}, \frac{\mathbf{1}}{n}\right) \right) - \sum_{t=1}^{T} \alpha_t \mathbf{p}^{*,\ell,\top} \mathbf{A}\mathbf{w}_t + \sum_{t=1}^{T} \alpha_t \mathbf{p}_t^\top \mathbf{A}(\mathbf{w}_t - \mathbf{w}_{t-1}) \\
&\quad - \sum_{t=1}^{T} \beta_t D_E\left(\mathbf{p}_t, \frac{\mathbf{1}}{n}\right) \\
&\overset{(b)}{\leq} \sum_{t=1}^{T} \alpha_t \mathbf{p}^{*,\ell,\top} \mathbf{A}(\mathbf{w}_{t-1} - \mathbf{w}_t) + \sum_{t=1}^{T} \alpha_t \mathbf{p}_t^\top \mathbf{A}(\mathbf{w}_t - \mathbf{w}_{t-1}) + 2\log n \sum_{t=1}^{T} \beta_t \\
&\overset{(c)}{\leq} \sum_{t=1}^{T} \alpha_t \|\mathbf{p}^{*,\ell}\|_1 \|\mathbf{A}(\mathbf{w}_{t-1} - \mathbf{w}_t)\|_\infty + \sum_{t=1}^{T} \alpha_t \|\mathbf{p}_t\|_1 \|\mathbf{A}(\mathbf{w}_t - \mathbf{w}_{t-1})\|_\infty + 2\log n \sum_{t=1}^{T} \beta_t \\
&\overset{(d)}{\leq} 2\sum_{t=1}^{T} \alpha_t \|\mathbf{A}(\mathbf{w}_{t-1} - \mathbf{w}_t)\|_\infty + 2\log n \sum_{t=1}^{T} \beta_t \\
&= 2\sum_{t=1}^{T} \alpha_t \underset{i\in[n]}{\text{argmax}} \left| y^{(i)} \mathbf{x}^{(i)\top}(\mathbf{w}_{t-1} - \mathbf{w}_t) \right| + 2\log n \sum_{t=1}^{T} \beta_t \\
&\overset{(e)}{\leq} 2\sum_{t=1}^{T} \alpha_t \|\mathbf{w}_{t-1} - \mathbf{w}_t\|_q + 2\log n \sum_{t=1}^{T} \beta_t \\
&= 2\sum_{t=2}^{T} \alpha_t \|\mathbf{w}_{t-1} - \mathbf{w}_t\|_q + 2\log n \sum_{t=1}^{T} \beta_t + \alpha_1 \|\mathbf{w}_1\|_q \\
&\overset{(f)}{\leq} 2\sum_{t=2}^{T} \frac{2\alpha_t^2}{2\sum_{j=1}^{t-1} \alpha_j(q-1)} + 2\sum_{t=2}^{T} \frac{\sum_{j=1}^{t-1} \alpha_j(q-1)}{2\cdot 2} \|\mathbf{w}_t - \mathbf{w}_{t-1}\|_q^2 + 2\log n \sum_{t=1}^{T} \beta_t + \alpha_1 \|\mathbf{w}_1\|_q \\
&= 2\sum_{t=2}^{T} \frac{\alpha_t^2}{\sum_{j=1}^{t-1} \alpha_j(q-1)} + \sum_{t=2}^{T} \frac{\sum_{j=1}^{t-1} \alpha_j(q-1)}{2} \|\mathbf{w}_t - \mathbf{w}_{t-1}\|_q^2 + 2\log n \sum_{t=1}^{T} \beta_t + \alpha_1 \|\mathbf{w}_1\|_q.
\end{aligned}
\tag{21}
$$

Here, inequality $(a)$ is based on the optimality of $\mathbf{p}_t$, inequality $(b)$ is due to the fact that the negative entropy regularizer is upper bounded, inequality $(c)$ is because of the Hölder's inequality, inequality $(d)$ is derived from $\mathbf{p}_t, \mathbf{p} \in \Delta^n$, inequality $(e)$ is based on the Hölder's inequality and $\|\mathbf{x}^{(i)}\|_p \leq 1$ for all $i \in [n]$, and inequality $(f)$ is based on Young's inequality:

$$
\sum_{t=2}^{T} \alpha_t \|\mathbf{w}_{t-1} - \mathbf{w}_t\|_q \leq \sum_{t=2}^{T} \frac{2\alpha_t^2}{2\sum_{j=1}^{t-1} \alpha_j(q-1)} + \sum_{t=2}^{T} \frac{\sum_{j=1}^{t-1} \alpha_j(q-1)}{2\cdot 2} \|\mathbf{w}_t - \mathbf{w}_{t-1}\|_q^2,
$$

where we pick $\frac{\sum_{j=1}^{t-1} \alpha_i(q-1)}{2}$ as the constant of Young's inequality. Finally, note that $\mathbf{w}_1 = [\nabla\Phi]^{-1}(\mathbf{A}^\top \mathbf{p}_1)$, so based on the property of $p$-norm, we have

$$
\alpha_1 \|\mathbf{w}_1\|_q = \alpha_1 \|[\nabla\Phi]^{-1}(\mathbf{A}^\top \mathbf{p}_1)\|_q = \alpha_1 \|\mathbf{A}^\top \mathbf{p}_1\|_p \leq \alpha_1.
$$

For the $\mathbf{w}$-player, note that $h_t(\mathbf{w})$ is $(q-1)$-strongly convex wrt the $\|\cdot\|_q$-norm. Thus, based on Lemma 3, we have

$$\text{Reg}_T^{\mathbf{w}} \le - \sum_{t=1}^{T} \left( \frac{(q-1)\sum_{s=1}^{t-1}\alpha_s}{2} \right) \|\mathbf{w}_t - \mathbf{w}_{t-1}\|_q^2. \tag{22}$$

## C.2 Proof of Corollary 1

Note that $C_T \le \frac{\gamma^2}{4}$ is equivalent to $\frac{\gamma^2}{4}\sum_{t=1}^{T}\alpha_t \ge (\text{Reg}_T^{\mathbf{P}} + \text{Reg}_T^{\mathbf{w}})$. We have $\frac{\gamma^2}{4}\sum_{t=1}^{T}\alpha_t = \frac{\gamma^2}{4}\sum_{t=1}^{T}\frac{1}{\sqrt{t}} \ge \frac{\gamma^2}{4}\sqrt{T}$, and

$$\begin{aligned}
\text{Reg}_T^{\mathbf{P}} + \text{Reg}_T^{\mathbf{w}} &= \left( 2\sum_{t=2}^{T}\frac{\alpha_t^2}{\sum_{j=1}^{t-1}\alpha_j(q-1)} + 2\log n \sum_{t=2}^{T}\frac{\alpha_t}{\sum_{i=1}^{t-1}\alpha_i} \right) + \alpha_1(1+2\log n) \\
&= 2\sum_{t=2}^{T}\frac{1/t}{(q-1)\sum_{j=1}^{t-1}1/\sqrt{j}} + 2\log n \sum_{t=2}^{T}\frac{1/\sqrt{t}}{\sum_{i=1}^{t-1}1/\sqrt{j}} + 1 + 2\log n \\
&\le 2\sum_{t=2}^{T}\frac{1}{(q-1)t\sqrt{t-1}} + \log n \sum_{t=2}^{T}\frac{1}{\sqrt{t}\sqrt{t-1}} + 1 + 2\log n \\
&\le \frac{2}{q-1}(\log T+1) + \log n(\log T+1) + 1 + 2\log n \\
&\le \left( \frac{2}{q-1} + \log n \right)\log T + 4\log n + \frac{4}{q-1}.
\end{aligned} \tag{23}$$

Therefore, $C_T \le \frac{\gamma^2}{4}$ when

$$\sqrt{T} \ge \frac{4\left( \left( \frac{2}{q-1} + \log n \right)\log T + 4\log n + \frac{4}{q-1} \right)}{\gamma^2},$$

and in this case

$$\frac{\min_{\mathbf{p}\in\Delta^n}\mathbf{p}^\top \mathbf{A}\widetilde{\mathbf{v}}_T}{\|\widetilde{\mathbf{v}}_T\|_q} \ge \gamma - \frac{4\left( \left( \frac{2}{q-1} + \log n \right)\log T + 4\log n + \frac{4}{q-1} \right)}{\gamma^2\sqrt{T}}. \tag{24}$$

## C.3 Proof of Corollary 2

We have $\frac{\gamma^2}{4}\sum_{t=1}^{T}\alpha_t = \frac{\gamma^2}{4}T$, and

$$\begin{aligned}
\text{Reg}_T^{\mathbf{P}} + \text{Reg}_T^{\mathbf{w}} &= \left( 2\sum_{t=2}^{T}\frac{\alpha_t^2}{\sum_{j=1}^{t-1}\alpha_j(q-1)} + 2\log n \sum_{t=2}^{T}\frac{\alpha_t}{\sum_{i=1}^{t-1}\alpha_i} \right) + \alpha_1(1+2\log n) \\
&= 2\sum_{t=2}^{T}\frac{1}{(t-1)(q-1)} + 2\log n \sum_{t=2}^{T}\frac{1}{t-1} + 1 + 2\log n \\
&= 1 + 2\log n + \left( \frac{2}{q-1} + 2\log n \right)\sum_{t=1}^{T-1}\frac{1}{t} \\
&\le \left( \frac{2}{q-1} + 2\log n \right)(\log T+2).
\end{aligned} \tag{25}$$

Therefore, $C_T \le \frac{\gamma^2}{4}$ when

$$T \ge \frac{4\left( \frac{2}{q-1} + 2\log n \right)(\log T+2)}{\gamma^2},$$

and in this case

$$\frac{\min_{\mathbf{p}\in\Delta^n} \mathbf{p}^\top \mathbf{A}\widetilde{\mathbf{v}}_T}{\|\widetilde{\mathbf{v}}_T\|_q} \geq \gamma - \frac{4\left(\frac{2}{q-1} + 2\log n\right)(\log T + 2)}{\gamma^2 T}. \tag{26}$$

and in this case

## C.4 Proof of Corollary 3

We have $\frac{\gamma^2}{4}\sum_{t=1}^T \alpha_t = \frac{\gamma^2}{4}\frac{T(T+1)}{2} \geq \frac{\gamma^2}{8}T^2$, and

$$\left(2\sum_{t=2}^T \frac{\alpha_t^2}{\sum_{j=1}^{t-1}\alpha_j(q-1)} + 2\log n\sum_{t=2}^T \frac{\alpha_t}{\sum_{i=1}^{t-1}\alpha_i}\right) + \alpha_1(1 + 2\log n)$$

$$= 2\sum_{t=2}^T \frac{2t^2}{t(t-1)(q-1)} + 2\log n\sum_{t=2}^T \frac{2t}{t(t-1)} + 1 + 2\log n \tag{27}$$

$$\leq \frac{8T}{q-1} + 4\log n\log T + 1 + 2\log n.$$

Therefore, $C_T \leq \frac{\gamma^2}{4}$ when

$$T \geq \sqrt{\frac{8\left[\frac{8T}{q-1} + 4\log n\log T + 1 + 2\log n\right]}{\gamma^2}},$$

and in this case

$$\frac{\min_{\mathbf{p}\in\Delta^n} \mathbf{p}^\top \mathbf{A}\widetilde{\mathbf{v}}_T}{\|\widetilde{\mathbf{v}}_T\|_q} \geq \gamma - \frac{32}{\gamma^2 T(q-1)} - \frac{8(4\log n\log T + 1 + 2\log n)}{\gamma^2 T^2}. \tag{28}$$

## C.5 Proof of Theorem 4

We first focus on the algorithm equivalence. We have

$$\mathbf{w}_t = \underset{\mathbf{w}\in\mathbb{R}^d}{\operatorname{argmin}} -\sum_{i=1}^{t-1}\alpha_i\mathbf{p}_i^\top \mathbf{A}\mathbf{w} - \alpha_t\mathbf{p}_{t-1}^\top \mathbf{A}\mathbf{w} + \sum_{i=1}^t \alpha_i\Phi(\mathbf{w})$$

$$= [\nabla\Phi]^{-1}\left(\frac{1}{\sum_{i=1}^t \alpha_i}\left(\sum_{i=1}^{t-1}\alpha_i\mathbf{A}^\top\mathbf{p}_i + \alpha_t\mathbf{A}^\top\mathbf{p}_{t-1}\right)\right), \tag{29}$$

So we have

$$\nabla\Phi(\mathbf{w}_t) = \frac{1}{\sum_{i=1}^t \alpha_i}\left(\sum_{i=1}^{t-1}\alpha_i\mathbf{A}^\top\mathbf{p}_i + \alpha_t\mathbf{A}^\top\mathbf{p}_{t-1}\right)$$

$$= \frac{1}{\sum_{i=1}^t \alpha_i}\left(\frac{\sum_{i=1}^{t-1}\alpha_i}{\sum_{i=1}^{t-1}\alpha_i}\left(\sum_{i=1}^{t-2}\alpha_i\mathbf{A}^\top\mathbf{p}_i + \alpha_{t-1}\mathbf{A}^\top\mathbf{p}_{t-2}\right) + \alpha_t\mathbf{A}^\top\mathbf{p}_{t-1} + \alpha_{t-1}\mathbf{A}^\top(\mathbf{p}_{t-1} - \mathbf{p}_{t-2})\right)$$

$$= \frac{1}{\sum_{i=1}^t \alpha_i}\left(\left[\sum_{i=1}^{t-1}\alpha_i\right]\nabla\Phi(\mathbf{w}_{t-1}) + \alpha_t\mathbf{A}^\top\mathbf{p}_{t-1} + \alpha_{t-1}\mathbf{A}^\top(\mathbf{p}_{t-1} - \mathbf{p}_{t-2})\right). \tag{30}$$

Therefore, we have

$$\nabla\Phi\left(\mathbf{w}_t\sum_{i=1}^t \alpha_i\right) = \nabla\Phi\left(\mathbf{w}_{t-1}\sum_{i=1}^{t-1}\alpha_i\right) + \alpha_t\mathbf{A}^\top\mathbf{p}_{t-1} + \alpha_{t-1}\mathbf{A}^\top(\mathbf{p}_{t-1} - \mathbf{p}_{t-2}). \tag{31}$$

On the other hand, for the **p**-player, we have

$$\mathbf{p}_t = \operatorname*{argmin}_{\mathbf{p} \in \Delta^n} \alpha_t \ell_t(\mathbf{p}) + \beta_t D_{\text{KL}}\left(\mathbf{p}, \frac{\mathbf{1}}{n}\right).$$

Based on the relationship between OMD with the negative entropy regularizer on the simplex (Hazan, 2016), it is easy to verify that $\forall i \in [n], t \in [T]$,

$$p_{t,i} = \frac{\exp(-\frac{\alpha_t}{\beta_t} y^{(i)} \mathbf{x}^{(i)\top} \mathbf{w}_t)}{\sum_{j=1}^{n} \exp(-\frac{\alpha_t}{\beta_t} y^{(j)} \mathbf{x}^{(j)\top} \mathbf{w}_t)},$$

which implies that

$$\mathbf{A}^\top \mathbf{p}_t = -\frac{\nabla L\left(\frac{\alpha_t}{\beta_t} \mathbf{w}_t\right)}{L\left(\frac{\alpha_t}{\beta_t} \mathbf{w}_t\right)}.$$

Combining the equations above and replace $\sum_{k=1}^{t} \alpha_k \mathbf{w}_k$ with $\mathbf{v}_t$, we get

$$\nabla \Phi(\mathbf{v}_t) = \nabla \Phi(\mathbf{v}_{t-1}) - \alpha_t \frac{\nabla L(\mathbf{v}_{t-1})}{L(\mathbf{v}_{t-1})} - \alpha_{t-1}\left(\frac{\nabla L(\mathbf{v}_{t-1})}{L(\mathbf{v}_{t-1})} - \frac{\nabla L(\mathbf{v}_{t-2})}{L(\mathbf{v}_{t-2})}\right).$$

The proof is finished by setting $\alpha_t = t$.

Next, we focus on the regret. For the **w**-player, Note that $h_t(\mathbf{w})$ is $(q-1)$-strongly convex wrt the $\|\cdot\|$-norm. Let $\widehat{\mathbf{w}}_t = \operatorname*{argmin}_{\mathbf{w} \in \mathbb{R}^d} \sum_{i=1}^{t-1} \alpha_i h_i(\mathbf{w})$. Then, based on Lemma 4, we have

$$
\begin{aligned}
\text{Reg}_T^{\mathbf{w}} &\leq \sum_{t=1}^{T} \alpha_t \left(h_t(\mathbf{w}_t) - h_t(\widehat{\mathbf{w}}_{t+1}) - h_{t-1}(\mathbf{w}_t) + h_{t-1}(\widehat{\mathbf{w}}_{t+1})\right) \\
&\quad - \sum_{t=1}^{T} \frac{\sum_{i=1}^{t} \alpha_i(q-1)}{2} \|\mathbf{w}_t - \widehat{\mathbf{w}}_{t+1}\|_q^2 \\
&= \sum_{t=1}^{T} \alpha_t (\mathbf{p}_t - \mathbf{p}_{t-1})^\top \mathbf{A}(\mathbf{w}_t - \widehat{\mathbf{w}}_{t+1}) - \sum_{t=1}^{T} \frac{\sum_{i=1}^{t} \alpha_i(q-1)}{2} \|\mathbf{w}_t - \widehat{\mathbf{w}}_{t+1}\|_q^2 \\
&\leq \sum_{t=1}^{T} \alpha_t \|(\mathbf{p}_t - \mathbf{p}_{t-1})^\top \mathbf{A}\|_p \|\mathbf{w}_t - \widehat{\mathbf{w}}_{t+1}\|_q - \sum_{t=1}^{T} \frac{\sum_{i=1}^{t} \alpha_i(q-1)}{2} \|\mathbf{w}_t - \widehat{\mathbf{w}}_{t+1}\|_q^2 \\
&\leq \sum_{t=1}^{T} \frac{\alpha_t^2}{2 \sum_{i=1}^{t} \alpha_i(q-1)} \|\mathbf{A}^\top(\mathbf{p}_t - \mathbf{p}_{t-1})\|_p^2 + \frac{\sum_{i=1}^{t} \alpha_i(q-1)}{2} \|\mathbf{w}_t - \widehat{\mathbf{w}}_{t+1}\|_q^2 \\
&\quad - \sum_{t=1}^{T} \frac{\sum_{i=1}^{t} \alpha_i(q-1)}{2} \|\mathbf{w}_t - \widehat{\mathbf{w}}_{t+1}\|_q^2 \\
&= \frac{1}{2(q-1)} \sum_{t=1}^{T} \frac{\alpha_t^2}{\sum_{i=1}^{t} \alpha_i} \left\|\sum_{i=1}^{n} (p_{t,i} - p_{t-1,i}) y^{(i)} \mathbf{x}^{(i)}\right\|_p^2 \\
&\leq \frac{1}{2(q-1)} \sum_{t=1}^{T} \frac{\alpha_t^2}{\sum_{i=1}^{t} \alpha_i} \|\mathbf{p}_t - \mathbf{p}_{t-1}\|_1^2,
\end{aligned}
\tag{32}
$$

where the first inequality is based on Hölder's inequality, the second inequality is based on Young's inequality, i.e.,

$$
\begin{aligned}
&\sum_{t=1}^{T} \alpha_t \|(\mathbf{p}_t - \mathbf{p}_{t-1})^\top \mathbf{A}\|_p \|\mathbf{w}_t - \widehat{\mathbf{w}}_{t+1}\|_q \\
&\leq \sum_{t=1}^{T} \frac{\alpha_t^2}{2 \sum_{i=1}^{t} \alpha_i(q-1)} \|\mathbf{A}^\top(\mathbf{p}_t - \mathbf{p}_{t-1})\|_p^2 + \frac{\sum_{i=1}^{t} \alpha_i(q-1)}{2} \|\mathbf{w}_t - \widehat{\mathbf{w}}_{t+1}\|_q^2
\end{aligned}
\tag{33}
$$

For the **p**-player, we have

$$\sum_{t=1}^{T} \alpha_t \mathbf{p}_t^T \mathbf{A} \mathbf{w}_t - \min_{\mathbf{p}\in\Delta^n} \alpha_t \mathbf{p}^\top \mathbf{A} \mathbf{w}_t$$

$$= \sum_{t=1}^{T} (\alpha_t \mathbf{p}_t^\top \mathbf{A} \mathbf{w}_t + \beta_t D_{\mathrm{KL}}\left(\mathbf{p}_t, \frac{\mathbf{1}}{n}\right) - \min_{\mathbf{p}\in\Delta^n} \sum_{t=1}^{T} \alpha_t \mathbf{p}^\top \mathbf{A} \mathbf{w}_t - \sum_{t=1}^{T} \beta_t D_E\left(\mathbf{p}_t, \frac{\mathbf{1}}{n}\right) \qquad (34)$$

$$\le 2\sum_{t=1}^{T} \beta_t \log n = 2 \sum_{t=1}^{T} \frac{\alpha_t}{\sum_{i=1}^{t} \alpha_i} \log n.$$

Finally, we focus on the margin and implicit bias. Since $\alpha_t = t$, for the **w**-player's regret, we have

$$\frac{1}{2(q-1)} \sum_{t=1}^{T} \frac{\alpha_t^2}{\sum_{i=1}^{t} \alpha_i} \|\mathbf{p}_t - \mathbf{p}_{t-1}\|_1^2 \le \frac{1}{(q-1)} \sum_{t=1}^{T} \|\mathbf{p}_t - \mathbf{p}_{t-1}\|_1^2,$$

and for the **p**-player, we have

$$\sum_{t=1}^{T} \frac{\alpha_t}{\sum_{i=1}^{t} \alpha_i} \log n \le 4 \log T \log n.$$

Following Theorem 1, we have the margin and implicit bounds when

$$\sum_{t=1}^{T} \alpha_t = \frac{T(T+1)}{2} \ge \frac{T^2}{2} \ge \frac{4}{\gamma^2}\left(4\log n \log T + \frac{2T}{(q-1)}\right)$$

$$\ge \frac{4}{\gamma^2}\left(4\log n \log T + \frac{1}{(q-1)} \sum_{t=1}^{T} \|\mathbf{p}_t - \mathbf{p}_{t-1}\|_1^2\right), \qquad (35)$$

since the RHS is exactly $\frac{\gamma^2}{4}(\mathrm{Reg}_T^{\mathbf{p}} + \mathrm{Reg}_T^{\mathbf{w}})$.

## D  Omitted Proof in Section 4.2

In this section, we provide the proof related to the steepest descent algorithm. We first restate Algorithm 3, which is presented in Algorithm 6. Here, we provide two online dynamic under the game framework. They both are equivalent to the steepest descent algorithm in the left box, in the sense that $\mathbf{v}_T = \widetilde{\mathbf{w}}_T$. The left one is good for recovering the results in Nacson et al. (2019), while the bottom one is more suitable for analysing our accelerated rates.

We first recover the results of Nacson et al. (2019) in Theorem 8, and then proof our Theorem 5.

---

**Algorithm 6** Steepest Descent [Recall $\ell_t(\mathbf{p}) = g(\mathbf{p}, \mathbf{w}_t)$, and $h_t(\mathbf{w}) = -g(\mathbf{p}_t, \mathbf{w})$]

| |
|---|
| 1: **for** $t = 1, \ldots, T$ **do**   ||   **p**-player: $\mathbf{p}_t = \operatorname*{argmin}_{\mathbf{p}\in\Delta^n} \sum_{i=1}^{t-1} \alpha_i \ell_i(\mathbf{p}) + D_{\mathrm{KL}}\left(\mathbf{p}, \frac{1}{n}\right)$ |

1: **for** $t = 1, \ldots, T$ **do**   ||  **p**-player: $\mathbf{p}_t = \operatorname*{argmin}_{\mathbf{p}\in\Delta^n} \sum_{i=1}^{t-1} \alpha_i \ell_i(\mathbf{p}) + D_{\mathrm{KL}}\left(\mathbf{p}, \frac{1}{n}\right)$

2:    $\mathbf{s}_{t-1} = \operatorname*{argmin}_{\|\mathbf{s}\|\le 1} \mathbf{s}^\top \nabla L(\mathbf{v}_{t-1})$

3:    $\mathbf{v}_t = \mathbf{v}_{t-1} + \eta_{t-1}\mathbf{s}_{t-1}$  ||  **w**-player: $\mathbf{w}_t = \operatorname*{argmin}_{\mathbf{w}\in\mathbb{R}^d} \alpha_t h_t(\mathbf{w})$

4: **end for**

5: **Output**: $\mathbf{v}_T$  ||  **Output**: $\widetilde{\mathbf{w}}_T = \sum_{t=1}^{T} \alpha_t \mathbf{w}_t$

---

**w**-player: $\mathbf{w}_t = \operatorname*{argmin}_{\mathbf{w}\in\mathbb{R}^d} \langle \delta_{t-1}\alpha_{t-1}\nabla h_{t-1}(\mathbf{w}_{t-1}), \mathbf{w}\rangle + D_{\frac{1}{2}\|\cdot\|^2}(\mathbf{w}, \mathbf{w}_{t-1})$

**p**-player: $\mathbf{p}_t = \operatorname*{argmin}_{\mathbf{p}\in\Delta^n} \sum_{i=1}^{t} \alpha_i \ell_i(\mathbf{p}) + D_{\mathrm{KL}}\left(\mathbf{p}, \frac{1}{n}\right)$

**Output**: $\widetilde{\mathbf{w}}_T = \sum_{t=1}^{T} \alpha_t \mathbf{w}_t$.

---

**Theorem 8.** *Suppose Assumption 1 holds wrt a general norm $\|\cdot\|$. Let $\eta_t = \frac{\alpha_t \|\nabla L(\mathbf{w}_{t-1})\|_*}{L(\mathbf{w}_{t-1})}$.*
*Then, the methods in the top two boxes of Algorithm 6 are equivalent, in the sense that $\mathbf{v}_T = \widetilde{\mathbf{w}}_T$.*
*Moreover, let $\alpha_t = \frac{1}{\sqrt{t}}$. Then $C_T = \frac{\log n + 2\log T + 2}{\sqrt{T}}$. Therefore, when $T$ is sufficiently large such that*
$\sqrt{T} \geq \frac{4(\log n + 2\log T + 2)}{\gamma^2}$, *we have*

$$\frac{\min_{\mathbf{p} \in \Delta^n} \mathbf{p}^\top \mathbf{A} \widetilde{\mathbf{w}}_T}{\|\widetilde{\mathbf{w}}_T\|} \geq \gamma - \frac{4(2 + \log n + 2\log T)}{\gamma^2 \sqrt{T}}. \tag{36}$$

The first part of Theorem 8 shows that the steepest descent algorithm can be seen as an online learning dynamic where the in each round $\mathbf{p}$-player performs the standard *follow-the-regularized-leader* algorithm (Hazan, 2016), while the $\mathbf{w}$-player uses *best-response$^+$* (Wang et al., 2021b), meaning it picks the decision by directly minimizing the loss of the round. The second part of the theorem shows that the margin convergence rate, which recovers the $\gamma - O\left(\frac{\log n + \log T}{\sqrt{T}}\right)$ rate of Nacson et al. (2019), while the convergence in terms of distance is new. Note that the strongly convex condition is not required for the margin maximization analysis, but is needed for the distance convergence analysis. Next, we restate Theorem 5, where an improved margin maximization rate when the strong convexity condition is met.

**Theorem 9** (Restate of Theorem 5)**.** *Suppose Assumption 1 holds wrt a general norm $\|\cdot\|$, and $\frac{1}{2}\|\cdot\|^2$ is $\lambda$-strongly convex wrt $\|\cdot\|$. Let $\eta_t = \frac{\alpha_t \|\nabla L(\mathbf{w}_t)\|}{L(\mathbf{w}_t)}$. Then the methods in the first and third boxes of Algorithm 6 are are equivalent, in the sense that $\mathbf{v}_T = \widetilde{\mathbf{w}}_T$. Moreover, let $\alpha_t = \frac{\lambda}{2}$. Then $C_T = \frac{\frac{\lambda}{4} + \log n}{T\lambda}$. Therefore, when $T \geq \frac{\lambda + 4\log n}{\lambda \gamma^2}$, we have*

$$\frac{\min_{\mathbf{p} \in \Delta^n} \mathbf{p}^\top \mathbf{A} \mathbf{v}_T}{\|\mathbf{v}_T\|} \geq \gamma - \frac{\lambda + 4\log n}{\gamma^2 T\lambda}. \text{ and } \left\|\frac{\mathbf{v}_T}{\|\mathbf{v}_T\|} - \mathbf{w}^*_{\|\cdot\|}\right\|^2 \leq \frac{8}{\sqrt{\lambda}\gamma^2}\sqrt{\frac{\lambda + 2\log n}{T\lambda}}.$$

### D.1 Proof of Theorem 8

We first focus on the algorithm equivalence between the top two boxes. Based on the relationship between FTRL and EWA (Orabona, 2019), one can verify that

$$p_{t,i} = \frac{\exp(-y^{(i)}\mathbf{x}^{(i)\top}(\sum_{k=1}^{t-1} \alpha_k \mathbf{w}_k))}{\sum_{j=1}^n \exp(-y^{(j)}\mathbf{x}^{(j)\top}(\sum_{k=1}^{t-1} \alpha_k \mathbf{w}_k))}.$$

Combining with the definition of $L$ and $\widetilde{\mathbf{w}}_t$, it implies that

$$\frac{\nabla L(\widetilde{\mathbf{w}}_{t-1})}{L(\widetilde{\mathbf{w}}_{t-1})} = -\mathbf{A}^\top \mathbf{p}_t,$$

That is, $\nabla L(\widetilde{\mathbf{w}}_{t-1}) = -L(\widetilde{\mathbf{w}}_{t-1})\mathbf{A}^\top \mathbf{p}_t$. Let $\widetilde{\mathbf{s}}_t = \operatorname{argmin}_{\|\mathbf{s}\| \leq 1} \mathbf{s}^\top \frac{\nabla L(\widetilde{\mathbf{w}}_{t-1})}{L(\widetilde{\mathbf{w}}_{t-1})}$. Note that, on one hand, we have

$$\widetilde{\mathbf{s}}_t = \operatorname*{argmin}_{\|\mathbf{s}\| \leq 1} \mathbf{s}^\top \frac{\nabla L(\widetilde{\mathbf{w}}_{t-1})}{L(\widetilde{\mathbf{w}}_{t-1})} = \operatorname*{argmin}_{\|\mathbf{s}\| \leq 1} \mathbf{s}^\top \nabla L(\widetilde{\mathbf{w}}_{t-1}), \tag{37}$$

where the second equality is because because the *argmin* does not change if we scale the objective functions. On the other hand, we have

$$\widetilde{\mathbf{s}}_t = \operatorname*{argmin}_{\|\mathbf{s}\| \leq 1} \mathbf{s}^\top \frac{\nabla L(\widetilde{\mathbf{w}}_{t-1})}{L(\widetilde{\mathbf{w}}_{t-1})} = \operatorname*{argmax}_{\|\mathbf{s}\| \leq 1} \mathbf{s}^\top \left(-\frac{\nabla L(\widetilde{\mathbf{w}}_{t-1})}{L(\widetilde{\mathbf{w}}_{t-1})}\right). \tag{38}$$

To proceed, we introduce the following lemma.

**Lemma 2.** *Let $\|\cdot\|$ be any norm in $\mathbb{R}^d$. Let $\mathbf{a} \in \mathbb{R}^d$, and*

$$\mathbf{s} = \operatorname*{argmax}_{\|\mathbf{s}'\| \leq 1} \mathbf{s}'^\top \mathbf{a}. \tag{39}$$

*Then*

$$\|\mathbf{a}\|_* \mathbf{s} = \operatorname*{argmin}_{\mathbf{x} \in \mathbb{R}^d} -\mathbf{a}^\top \mathbf{x} + \frac{1}{2}\|\mathbf{x}\|^2. \tag{40}$$

*Proof.* We first focus on (40). Note that the objective has two terms. For the first term, based on Hölder's inequality, we have

$$-\mathbf{a}^\top \mathbf{x} \geq -\|\mathbf{a}\|_* \|\mathbf{x}\|.$$

Let $\|\mathbf{x}\| = c$, where $c > 0$ is a constant, then the equality is achieved (and thus the first term of the objective function is minimized) when

$$\mathbf{x} = \operatorname*{argmin}_{\|\mathbf{x}'\| \leq c} -\mathbf{x}'^\top \mathbf{a} = \operatorname*{argmax}_{\|\mathbf{x}'\| \leq c} \mathbf{x}'^\top \mathbf{a}. \tag{41}$$

In this case, for the objective function of (40), we have $-\mathbf{a}^\top \mathbf{x} + \frac{1}{2}\|\mathbf{x}\|^2 = -c\|\mathbf{a}\|_* + \frac{1}{2}c^2$. It's easy to see that the objective function is minimized when $c = \|\mathbf{a}\|_*$. The proof is finished by combining (39) and (41). $\square$

This lemma shows that the best response direction (under our game) is the steepest direction. Thus, we have

$$\mathbf{w}_t = \operatorname*{argmin}_{\mathbf{w} \in \mathbb{R}^d} -\mathbf{p}_t^\top \mathbf{A}\mathbf{w} + \frac{1}{2}\|\mathbf{w}\|^2 = \operatorname*{argmin}_{\mathbf{w} \in \mathbb{R}^d} \mathbf{w}^\top \frac{\nabla L(\widetilde{\mathbf{w}}_{t-1})}{L(\widetilde{\mathbf{w}}_{t-1})} + \frac{1}{2}\|\mathbf{w}\|^2$$

$$= \operatorname*{argmin}_{\mathbf{w} \in \mathbb{R}^d} -\mathbf{w}^\top \left(-\frac{\nabla L(\widetilde{\mathbf{w}}_{t-1})}{L(\widetilde{\mathbf{w}}_{t-1})}\right) + \frac{1}{2}\|\mathbf{w}\|^2 \tag{42}$$

$$= \operatorname*{argmax}_{\|\mathbf{s}'\| \leq 1} \mathbf{w}^\top \left(-\frac{\nabla L(\widetilde{\mathbf{w}}_{t-1})}{L(\widetilde{\mathbf{w}}_{t-1})}\right) = \frac{\|\nabla L(\widetilde{\mathbf{w}}_{t-1})\|_*}{L(\widetilde{\mathbf{w}}_{t-1})} \widetilde{\mathbf{s}}_t.$$

where the first equality is based on the updete rule of $\mathbf{w}_t$ at the right box of Algorithm 6, the final inequality is based on Lemma 2 and (38). Note that, for the first equality, we dropped $\alpha_t$ as it is a scaling parameter and does not have a influence on *argmin*. Thus, combining with (37), we have

$$\mathbf{w}_t = \frac{\|\nabla L(\widetilde{\mathbf{w}}_{t-1})\|_*}{L(\widetilde{\mathbf{w}}_{t-1})}\widetilde{\mathbf{s}}_t = \frac{\|\nabla L(\widetilde{\mathbf{w}}_{t-1})\|_*}{(L(\widetilde{\mathbf{w}}_{t-1}))} \operatorname*{argmin}_{\|\mathbf{s}\| \leq 1} \mathbf{s}^\top \nabla L(\widetilde{\mathbf{w}}_{t-1}).$$

Finally, we have

$$\widetilde{\mathbf{w}}_t = \widetilde{\mathbf{w}}_{t-1} + \alpha_t \mathbf{w}_t = \widetilde{\mathbf{w}}_t + \alpha_t \frac{\|\nabla L(\widetilde{\mathbf{w}}_{t-1})\|_*}{L(\widetilde{\mathbf{w}}_{t-1})} \operatorname*{argmin}_{\|\mathbf{s}\| \leq 1} \mathbf{s}^\top \nabla L(\widetilde{\mathbf{w}}_{t-1}).$$

We can finish the first part of the proof by replacing $\widetilde{\mathbf{w}}_t$ with $\mathbf{v}_t$, $\alpha_t \frac{\|\nabla L(\widetilde{\mathbf{w}}_{t-1})\|_*}{L(\widetilde{\mathbf{w}}_{t-1})}$ with $\eta_t$, and noticing $\operatorname*{argmin}_{\|\mathbf{s}\| \leq 1} \mathbf{s}^\top \nabla L(\widetilde{\mathbf{w}}_{t-1})$ is $\mathbf{s}_{t-1}$.

Next, we study the regret bound. Note that the $\mathbf{p}$-player plays the FTRL algorithm on the simplex with a 1-strongly convex regularizer $D_{\mathrm{KL}}\left(\mathbf{p}, \frac{1}{n}\right)$ (wrt $\ell_1$-norm). Therefore, based on Lemma 5, the regret can be upper bounded by

$$\sum_{t=1}^{T} \alpha_t \ell_t(\mathbf{p}_t) - \sum_{t=1}^{T} \alpha_t \ell_t(\mathbf{p}^*) \leq \log n + 2 \sum_{t=1}^{T} \alpha_t^2 \|\mathbf{A}^\top \mathbf{w}_t\|_\infty^2$$

$$= \log n + 2 \sum_{t=1}^{T} \frac{1}{t} \left(\max_{i \in n} |y^{(i)}\mathbf{x}^{(i)\top}\mathbf{w}_t|\right)\left(\max_{i \in n} |y^{(i)}\mathbf{x}^{(i)\top}\mathbf{w}_t|\right)$$

$$\leq \log n + 2 \sum_{t=1}^{T} \frac{1}{t} \|\mathbf{w}_t\|^2$$

$$= \log n + 2 \sum_{t=1}^{T} \frac{1}{t} \|\mathbf{A}^\top \mathbf{p}_t\|_*^2 \leq \log n + 2 \log T + 2,$$

$$\tag{43}$$

where the equality is because

$$\|\mathbf{w}_t\| = \left\|\left\|\frac{\nabla L(\mathbf{w}_t)}{L(\mathbf{w}_t)}\right\|_* \widetilde{\mathbf{s}}_t\right\| = \left\|\frac{\nabla L(\mathbf{w}_t)}{L(\mathbf{w}_t)}\right\|_* = \|\mathbf{A}^\top \mathbf{p}_t\|_*.$$

On the other hand, the $\mathbf{w}$-player uses the best response$^+$ algorithm, and one can easily observe that the regret is upper bounded by 0. Finally, we have $\sum_{t=1}^{T} \alpha_t = \sum_{t=1}^{T} \frac{1}{\sqrt{t}} \geq \sqrt{T}$.

## D.2 Proof of Theorem 9

We first focus on the bottom box. For the $\mathbf{w}$-player, we show that, if we set $\delta_{t-1} = \frac{1}{\alpha_{t-1}}$, then this OMD algorithm is equivalent to the best response algorithm.

Specifically, note that $\Phi(\mathbf{w}) = \frac{1}{2}\|\mathbf{w}\|^2$ is now $\lambda$-strongly convex. Thus the corresponding mirror map is well-defined and unique, and the function $\nabla\Phi(\cdot)$ is invertible. Therefore, the solution for best response is

$$\widehat{\mathbf{w}}_t = \operatorname*{argmin}_{\mathbf{w}} \alpha_{t-1} h_{t-1}(\mathbf{w}) = \operatorname*{argmin}_{\mathbf{w}} h_{t-1}(\mathbf{w}) = \nabla\Phi^{-1}(\mathbf{A}^\top \mathbf{p}_{t-1}). \tag{44}$$

On the other hand, since the $\mathbf{w}$-player uses OMD, and the decision set is unbounded, we have

$$\nabla\Phi(\mathbf{w}_t) = \nabla\Phi(\mathbf{w}_{t-1}) - \delta_{t-1}\alpha_{t-1}\nabla h_{t-1}(\mathbf{w}_{t-1}) = \nabla\Phi(\mathbf{w}_{t-1}) + \mathbf{A}^\top \mathbf{p}_{t-1} - \nabla\Phi(\mathbf{w}_{t-1}) = \mathbf{A}^\top \mathbf{p}_{t-1}. \tag{45}$$

Note that $\delta_{t-1}\alpha_{t-1} = 1$. Combining (44) and (45), we can draw the conclusion that $\mathbf{w}_t$ and $\widehat{\mathbf{w}}_t$ are identical, which shows OMD (with $\delta_{t-1} = \frac{1}{\alpha_{t-1}}$) and BR (at round $t-1$) here are the same. We use the BR form the algorithm equivalence analysis, and OMD form for the regret analysis.

Next, we prove the algorithm equivalence of the left and bottom boxes in Algorithm 6. The arguments are similar to that in Appendix D.1. The difference is that the order of the $\mathbf{w}$-player and the $\mathbf{p}$-player is switched. Firstly, For the $\mathbf{p}$-player, based on the connection between FTRL and EWA, we have

$$p_{t,i} \propto \exp\left(-y^{(i)}\mathbf{x}^{(i)\top}\left(\sum_{j=1}^{t}\alpha_j\mathbf{w}_j\right)\right) = \exp\left(-y^{(i)}\mathbf{x}^{(i)\top}\widetilde{\mathbf{w}}_t\right).$$

Combining with the definition of $L$, it implies that

$$\frac{\nabla L(\widetilde{\mathbf{w}}_t)}{L(\widetilde{\mathbf{w}}_t)} = -\mathbf{A}^\top \mathbf{p}_t,$$

That is, $\nabla L(\widetilde{\mathbf{w}}_t) = -L(\widetilde{\mathbf{w}}_t)\mathbf{A}^\top \mathbf{p}_t$. Let

$$\widetilde{\mathbf{s}}_t = \operatorname*{argmax}_{\|\mathbf{s}\|\leq 1} -\mathbf{s}^\top \frac{\nabla L(\widetilde{\mathbf{w}}_t)}{L(\widetilde{\mathbf{w}}_t)} = \operatorname*{argmin}_{\|\mathbf{s}\|\leq 1} \mathbf{s}^\top \nabla L(\widetilde{\mathbf{w}}_t), \tag{46}$$

Combining the first equality in (46) and Lemma 2, we have

$$\frac{\|\nabla L(\widetilde{\mathbf{w}}_t)\|_*}{L(\widetilde{\mathbf{w}}_t)}\widetilde{\mathbf{s}}_t = \operatorname*{argmin}_{\mathbf{w}\in\mathbb{R}^d} \mathbf{w}^\top \frac{\nabla L(\widetilde{\mathbf{w}}_t)}{L(\widetilde{\mathbf{w}}_t)} + \frac{1}{2}\|\mathbf{w}\|^2 = \operatorname*{argmin}_{\mathbf{w}\in\mathbb{R}^d} -\mathbf{p}_t^\top \mathbf{A}\mathbf{w} + \frac{1}{2}\|\mathbf{w}\|^2 = \mathbf{w}_{t+1}.$$

Thus,

$$\mathbf{w}_t = \frac{\|\nabla L(\widetilde{\mathbf{w}}_{t-1})\|_*}{L(\widetilde{\mathbf{w}}_{t-1})}\widetilde{\mathbf{s}}_{t-1} = \frac{\|\nabla L(\widetilde{\mathbf{w}}_{t-1})\|_*}{(L(\widetilde{\mathbf{w}}_{t-1}))} \operatorname*{argmin}_{\|\mathbf{s}\|\leq 1} \mathbf{s}^\top \nabla L(\widetilde{\mathbf{w}}_{t-1}).$$

Finally, we have

$$\widetilde{\mathbf{w}}_t = \widetilde{\mathbf{w}}_{t-1} + \alpha_t\mathbf{w}_t = \widetilde{\mathbf{w}}_t + \alpha_t\frac{\|\nabla L(\widetilde{\mathbf{w}}_{t-1})\|_*}{L(\widetilde{\mathbf{w}}_{t-1})} \operatorname*{argmin}_{\|\mathbf{s}\|\leq 1} \mathbf{s}^\top \nabla L(\widetilde{\mathbf{w}}_{t-1}).$$

We can finish the first part of the proof by replacing $\widetilde{\mathbf{w}}_t$ with $\mathbf{v}_t$, $\alpha_t\frac{\|\nabla L(\widetilde{\mathbf{w}}_{t-1})\|_*}{L(\widetilde{\mathbf{w}}_{t-1})}$ with $\eta_{t-1}$, and $\operatorname*{argmin}_{\|\mathbf{s}\|\leq 1} \mathbf{s}^\top \nabla L(\widetilde{\mathbf{w}}_{t-1})$ with $\mathbf{s}_t$.

Next, we focus on regret. For the $\mathbf{w}$-player, it uses the OMD algorithm, and we set the initial point $\mathbf{w}_0 = \mathbf{0}$. Note that we fixed $\alpha_t = \frac{\lambda}{4}$ for all $t$, and thus step size $\delta_{t-1} = \frac{1}{\alpha_{t-1}} = \frac{4}{\lambda}$ is also fixed.

Therefore, Lemma 7 can be applied. Define $\mathbf{u} = \operatorname*{argmin}_{\mathbf{w} \in \mathbb{R}^d} \sum_{t=1}^{T} \alpha_t h_t(\mathbf{w})$, we have

$$
\begin{aligned}
\sum_{t=1}^{T} \alpha_t h_t(\mathbf{w}_t) - \min_{\mathbf{w} \in \mathbb{R}^d} \sum_{t=1}^{T} \alpha_t h_t(\mathbf{w}) &= \sum_{t=1}^{T} \alpha_t h_t(\mathbf{w}_t) - \sum_{t=1}^{T} \alpha_t h_t(\mathbf{u}) \\
&\leq \frac{\Phi(\mathbf{u})}{\delta} + \sum_{t=1}^{T} \frac{\delta \alpha_t^2}{\lambda} \left\| \nabla h_t(\mathbf{w}_t) \right\|_*^2 \\
&= \frac{\Phi(\mathbf{u})}{\delta} + \sum_{t=1}^{T} \frac{\alpha_t^2 \delta}{\lambda} \left\| -\mathbf{A}^\top \mathbf{p}_t + \nabla \Phi(\mathbf{w}_t) \right\|_*^2 \\
&= \frac{\Phi(\mathbf{u})}{\delta} + \sum_{t=1}^{T} \frac{\alpha_t^2 \delta}{\lambda} \left\| -\mathbf{A}^\top \mathbf{p}_t + \mathbf{A}^\top \mathbf{p}_{t-1} \right\|_*^2 \\
&= \alpha_T \Phi(\mathbf{u}) + \sum_{t=1}^{T} \frac{\alpha_t}{\lambda} \left\| -\mathbf{A}^\top \mathbf{p}_t + \mathbf{A}^\top \mathbf{p}_{t-1} \right\|_*^2 \\
&= \alpha_T \Phi(\mathbf{u}) + \sum_{t=1}^{T} \frac{\alpha_t}{\lambda} \left\| \sum_{i=1}^{n} y^{(i)} \mathbf{x}^{(i)} (p_{t,i} - p_{t-1,i}) \right\|_*^2 \\
&\leq \alpha_T \Phi(\mathbf{u}) + \sum_{t=1}^{T} \frac{\alpha_t}{\lambda} \left( \sum_{i=1}^{n} |p_{t,i} - p_{t-1,i}| \right)^2 \\
&\leq \alpha_T \Phi(\mathbf{u}) + \sum_{t=1}^{T} \frac{\alpha_t}{\lambda} \left\| \mathbf{p}_t - \mathbf{p}_{t-1} \right\|_1^2,
\end{aligned}
\tag{47}
$$

where the second-to-last inequality is derived using triangle inequality and the assumption that $\|\mathbf{x}^{(i)}\|_*$ is upper bounded by 1. Next, for $\mathbf{u}$, note that

$$
\begin{aligned}
\operatorname*{argmin}_{\mathbf{w} \in \mathbb{R}^d} \sum_{t=1}^{T} \alpha_t h_t(\mathbf{w}) &= \operatorname*{argmin}_{\mathbf{w} \in \mathbb{R}^d} - \sum_{t=1}^{T} \alpha_t \mathbf{p}_t^\top \mathbf{A} \mathbf{w} + \frac{\sum_{t=1}^{T} \alpha_t}{2} \|\mathbf{w}\|^2 \\
&= \operatorname*{argmin}_{\mathbf{w} \in \mathbb{R}^d} - \frac{1}{\sum_{t=1}^{T} \alpha_t} \sum_{t=1}^{T} \alpha_t \mathbf{p}_t^\top \mathbf{A} \mathbf{w} + \frac{1}{2} \|\mathbf{w}\|^2.
\end{aligned}
\tag{48}
$$

Based on Lemma 2, we have

$$
\mathbf{u} = \left\| \mathbf{A}^\top \left( \frac{1}{\sum_{t=1}^{T} \alpha_t} \sum_{t=1}^{T} \alpha_t \mathbf{p}_t \right) \right\|_* \mathbf{s},
$$

where

$$
\mathbf{s} = - \operatorname*{argmax}_{\|\mathbf{s}\| \leq 1} \mathbf{s}^\top \left( \mathbf{A}^\top \left( \frac{1}{\sum_{t=1}^{T} \alpha_t} \sum_{t=1}^{T} \alpha_t \mathbf{p}_t \right) \right).
$$

Therefore,

$$
\Phi(\mathbf{u}) = \frac{1}{2} \|\mathbf{u}\|^2 = \frac{1}{2} \left\| \mathbf{A}^\top \left( \frac{1}{\sum_{t=1}^{T} \alpha_t} \sum_{t=1}^{T} \alpha_t \mathbf{p}_t \right) \right\|_*^2 \leq \frac{1}{2},
$$

where the inequality is based on the triangle-inequality and the assumption that the dual norm of data is upper bounded by 1. Finally, for the $\mathbf{p}$-player, since it uses FTRL$^+$, based on Lemma 6, we have

$$
\sum_{t=1}^{T} \alpha_t \ell_t(\mathbf{p}_t) - \sum_{t=1}^{T} \ell_t(\mathbf{p}^*) \leq \log n - \sum_{t=1}^{T} \frac{1}{2} \|\mathbf{p}_t - \mathbf{p}_{t-1}\|_1^2.
$$

To summarize, and let $\alpha_t = \frac{\lambda}{2}$, we have

$$
\frac{\operatorname{Reg}_T^{\mathbf{w}} + \operatorname{Reg}_T^{\mathbf{P}}}{\sum_{t=1}^{T} \alpha_t} = \frac{\frac{\lambda}{4} + \log n}{T \lambda}.
$$

# E Proof of Theorem 6

We first focus on the equivalence between the left and bottom boxes of Algorithm 4. For the $\mathbf{w}$-player, similar to the proof of Theorem 7, we have

$$\mathbf{w}_t = \operatorname*{argmin}_{\mathbf{w} \in \mathbb{R}^d} \sum_{j=1}^t \alpha_j h_j(\mathbf{w}) = \operatorname*{argmin}_{\mathbf{w} \in \mathbb{R}^d} \sum_{j=1}^t -\alpha_j \mathbf{p}_j^\top \mathbf{A} \mathbf{w} + \frac{\sum_{j=1}^t \alpha_j}{2} \|\mathbf{w}\|^2$$

$$= [\nabla \Phi]^{-1} \left( \frac{1}{\sum_{j=1}^t \alpha_j} \sum_{j=1}^t \alpha_j \mathbf{A}^\top \mathbf{p}_j \right), \tag{49}$$

which implies that

$$\nabla \Phi(\mathbf{w}_t) = \frac{\sum_{j=1}^{t-1} \alpha_j}{\sum_{j=1}^t \alpha_j} \nabla \Phi(\mathbf{w}_{t-1}) + \frac{\alpha_t}{\sum_{j=1}^t \alpha_j} \mathbf{A}^\top \mathbf{p}_t,$$

and thus

$$\nabla \Phi \left( \mathbf{w}_t \sum_{j=1}^t \alpha_j \right) = \nabla \Phi \left( \mathbf{w}_{t-1} \sum_{j=1}^{t-1} \alpha_j \right) + \alpha_t \mathbf{A}^\top \mathbf{p}_t.$$

On the other hand, for the $\mathbf{p}$-player, since it performs Optimistic FTRL on a simplex with the negative entropy regularizer, we have

$$p_{t,i} \propto \exp \left( -c \left( \sum_{i=1}^{t-1} \alpha_i \mathbf{w}_i + \alpha_t \mathbf{w}_{t-1} \right)^\top \mathbf{x}^{(i)} y^{(i)} \right)$$

$$= \exp \left( -c \left( \widetilde{\mathbf{w}}_{t-1} + \alpha_t \mathbf{w}_{t-1} \right)^\top \mathbf{x}^{(i)} y^{(i)} \right), \tag{50}$$

which implies that

$$\frac{\nabla L(c \widetilde{\mathbf{w}}_{t-1} + c \alpha_t \mathbf{w}_{t-1})}{L(c \widetilde{\mathbf{w}}_{t-1} + c \alpha_t \mathbf{w}_{t-1})} = -\mathbf{A}^\top \mathbf{p}_t.$$

Let $\mathbf{z}_t = \mathbf{w}_t \sum_{i=1}^t \alpha_i$, then we have

$$\nabla \Phi (\mathbf{z}_t) = \nabla \Phi (\mathbf{z}_{t-1}) - \alpha_t \frac{\nabla L(c \widetilde{\mathbf{w}}_{t-1} + c \alpha_t \mathbf{w}_{t-1})}{L(c \widetilde{\mathbf{w}}_{t-1} + c \alpha_t \mathbf{w}_{t-1})} = \nabla \Phi (\mathbf{z}_{t-1}) - \alpha_t \frac{\nabla L(c \widetilde{\mathbf{w}}_{t-1} + \frac{c \alpha_t}{\sum_{i=1}^{t-1} \alpha_i} \mathbf{z}_{t-1})}{L(c \widetilde{\mathbf{w}}_{t-1} + \frac{c \alpha_t}{\sum_{i=1}^{t-1} \alpha_i} \mathbf{z}_{t-1})}.$$

Finally, notice that $\widetilde{\mathbf{w}}_t = \widetilde{\mathbf{w}}_{t-1} + \alpha_t \mathbf{w}_t = \widetilde{\mathbf{w}}_{t-1} + \frac{\alpha_t}{\sum_{i=1}^t \alpha_i} \mathbf{z}_t$. The proof is finished by replacing $\widetilde{\mathbf{w}}_t$ with $\widetilde{\mathbf{v}}_t$, $\mathbf{w}_t$ with $\frac{\mathbf{z}_t}{\sum_{i=1}^t \alpha_i}$, $c \widetilde{\mathbf{w}}_{t-1} + c \alpha_t \mathbf{w}_{t-1}$ with $\mathbf{v}_t$, configuring $\beta_{t,1} = c = \frac{\lambda}{4}$, $\beta_{t,1}' = \frac{c \alpha_t}{\sum_{i=1}^{t-1} \alpha_i} = \frac{\lambda}{2(t-1)}$, $\beta_{2,t} = 1$, $\beta_{2,t}' = \frac{\alpha_t}{\sum_{i=1}^t \alpha_t} = \frac{2}{t+1}$, $\eta_t = \frac{t}{L(\mathbf{v}_t)}$.

Next, we focus on the equivalence between the right and bottom boxes. Note that for the $\mathbf{w}$-player, we also have

$$\mathbf{w}_t = \operatorname*{argmin}_{\mathbf{w} \in \mathbb{R}^d} \sum_{j=1}^t \alpha_j h_j(\mathbf{w}) = \operatorname*{argmin}_{\mathbf{w} \in \mathbb{R}^d} \sum_{j=1}^t -\alpha_j \mathbf{p}_j^\top \mathbf{A} \mathbf{w} + \frac{\sum_{j=1}^t \alpha_j}{2} \|\mathbf{w}\|^2$$

$$= \operatorname*{argmin}_{\mathbf{w} \in \mathbb{R}^d} - \frac{1}{\sum_{j=1}^t \alpha_j} \sum_{j=1}^t \alpha_j \mathbf{p}_j^\top \mathbf{A} \mathbf{w} + \frac{1}{2} \|\mathbf{w}\|^2. \tag{51}$$

Let

$$\mathbf{s}_t = -\operatorname*{argmax}_{\|\mathbf{s}\| \le 1} \mathbf{s}^\top \left[ \frac{1}{\sum_{j=1}^t \alpha_j} \sum_{j=1}^t \alpha_j \mathbf{A}^\top \mathbf{p}_j \right].$$

Based on Lemma 2, we have

$$\left\| \frac{1}{\sum_{j=1}^t \alpha_j} \sum_{j=1}^t \alpha_j \mathbf{A}^\top \mathbf{p}_j \right\|_* \mathbf{s}_t = \mathbf{w}_t.$$

Next, let $\mathbf{g}_t = -\frac{1}{\sum_{j=1}^t \alpha_j} \sum_{j=1}^t \alpha_j \mathbf{A}^\top \mathbf{p}_j$, we know

$$\mathbf{g}_t = \frac{\sum_{j=1}^{t-1} \alpha_j}{\sum_{j=1}^t \alpha_j} \mathbf{g}_t + \left( -\frac{\alpha_t}{\sum_{j=1}^t \alpha_j} \mathbf{A}^\top \mathbf{p}_t \right).$$

For the $\mathbf{p}$-player, it is clear that due to the optimistic term, we have

$$-\mathbf{A}^\top \mathbf{p}_t = \frac{\nabla L(c\widetilde{\mathbf{w}}_{t-1} + c\alpha_t \mathbf{w}_{t-1})}{L(c\widetilde{\mathbf{w}}_{t-1} + c\alpha_t \mathbf{w}_{t-1})}.$$

To summarize, and let $\alpha_t = t$, we can conclude the proof by the following algorithm:

$$\mathbf{g}_t = \frac{t-1}{t+1} \mathbf{g}_{t-1} + \frac{2}{t+1} \frac{\nabla L(c\widetilde{\mathbf{w}}_{t-1} + ct\mathbf{w}_{t-1})}{L(c\widetilde{\mathbf{w}}_{t-1} + ct\mathbf{w}_{t-1})} = \frac{t-1}{t+1} \mathbf{g}_{t-1} + \frac{2}{t+1} \frac{\nabla L(c\widetilde{\mathbf{w}}_{t-1} + ct\|\mathbf{g}_{t-1}\|_* \mathbf{s}_{t-1})}{L(c\widetilde{\mathbf{w}}_{t-1} + ct\|\mathbf{g}_{t-1}\|_* \mathbf{s}_{t-1})},$$

$$\mathbf{s}_t = = -\operatorname*{argmax}_{\|\mathbf{s}\| \leq 1} -\mathbf{s}^\top \mathbf{g}_t = \operatorname*{argmin}_{\|\mathbf{s}\| \leq 1} \mathbf{s}^\top \mathbf{g}_t,$$

$$\widetilde{\mathbf{w}}_t = \widetilde{\mathbf{w}}_{t-1} + t\|\mathbf{g}_t\|_* \mathbf{s}_t,$$

and let $\beta_{t,3} = \frac{t-1}{t+1}$, $\beta_{t,4} = \frac{\lambda}{4}$, $\beta'_{t,4} = \frac{\lambda t \|\mathbf{g}_{t-1}\|_*}{4}$, $\beta'_{t,3} = \frac{2}{(t+1)L(\beta_{t,4}\mathbf{v}_{t-1} + \beta'_{t,4}\mathbf{s}_{t-1})}$, $\eta_t = t\|\mathbf{g}_t\|_*$.
Finally, we focus on the regret bound. For the $\mathbf{w}$-player, note that $\Phi(\mathbf{w})$ is $\lambda$-strongly convex with respect to $\|\cdot\|$. Thus, based on Lemma 3, we have

$$\sum_{t=1}^T \alpha_t h_t(\mathbf{w}_t) - \sum_{t=1}^T \alpha_t h_t(\mathbf{w}) \leq -\sum_{t=1}^T \frac{\lambda(t-1)}{4} \|\mathbf{w}_t - \mathbf{w}_{t-1}\|^2. \tag{52}$$

On the other hand, note that $c = \frac{\lambda}{4}$, so based on Lemma 8, we have

$$\sum_{t=1}^T \alpha_t \ell_t(\mathbf{p}_t) - \sum_{t=1}^T \alpha_t \ell_t(\mathbf{p}) \leq \frac{4\log n}{\lambda} + \frac{\lambda}{8} \sum_{t=1}^T t^2 \|\mathbf{A}\mathbf{w}_t - \mathbf{A}\mathbf{w}_{t-1}\|_\infty^2$$

$$= \frac{4\log n}{\lambda} + \frac{\lambda}{8} \sum_{t=1}^T t^2 \left( \max_{i \in [n]} |y^{(i)} \mathbf{x}^{(i)\top}(\mathbf{w}_t - \mathbf{w}_{t-1})| \right)^2 \tag{53}$$

$$\leq \frac{4\log n}{\lambda} + \frac{\lambda}{8} \sum_{t=1}^T t^2 \|\mathbf{w}_t - \mathbf{w}_{t-1}\|^2.$$

It is easy to verify that $\frac{t^2}{8} \leq \frac{t(t-1)}{4}$ for $t \geq 2$. So to summarize we get

$$C_T = \frac{8\log n}{\lambda T^2}.$$

The proof can be finished by plugging in Theorem 1.

## F    Regret Bounds for OCO Algorithms

In this section, we provide standard regret bounds for Follow-The-Leader[+] (FTL[+]), Optimistic Follow-The-Leader (OptimisticFTL), Follow-The-Regularized-Leader (FTRL), Follow-The-Regularized-Leader[+] (FTRL[+]), and Optimistic Follow-The-Regularized-Leader (Optimistic FTRL). Lemmas 3, 4, 5 and 6 are based on Lemma of 3 of Wang et al. (2021b), and Lemmas 7 and 8 comes from Theorems 6.8 and 7.35 of Orabona (2019).

**Lemma 3.** *Consider a weighted online learning problem with a series of $\lambda$-strongly functions $f_1, \ldots, f_T$, and a series of corresponding parameters $\alpha_1, \ldots, \alpha_T$. The $FTL^+$ algorithm, given by*

$$\mathbf{z}_t = \operatorname*{argmin}_{\mathbf{z} \in \mathbb{R}^d} \sum_{i=1}^t \alpha_i f_i(\mathbf{z}),$$

*achieves the following regret bound:*

$$\forall \mathbf{z} \in \mathbb{R}^d, \ \sum_{t=1}^T \alpha_t f_t(\mathbf{z}_t) - \sum_{t=1}^T \alpha_t f_t(\mathbf{z}) \leq -\sum_{t=1}^T \left( \frac{\lambda \sum_{s=1}^{t-1} \alpha_s}{2} \right) \|\mathbf{z}_t - \mathbf{z}_{t-1}\|^2.$$

**Lemma 4.** *Consider a weighted online learning problem with a series of $\lambda$-strongly functions $f_1, \ldots, f_T$, and a series of corresponding parameters $\alpha_1, \ldots, \alpha_T$. Let $\widehat{\mathbf{z}}_t = \arg\min_{\mathbf{z} \in \mathbb{R}^d} \sum_{i=1}^{t-1} \alpha_i h_i(\mathbf{z})$. Then, the Optimistic FTL algorithm, given by*

$$\mathbf{z}_t = \arg\min_{\mathbf{z} \in \mathbb{R}^d} \sum_{i=1}^{t-1} \alpha_i f_i(\mathbf{z}) + \alpha_t f_{t-1}(\mathbf{z}),$$

*achieves the following regret $\forall \mathbf{z} \in \mathbb{R}^d$:*

$$\sum_{t=1}^{T} \alpha_t f_t(\mathbf{z}_t) - \sum_{t=1}^{T} \alpha_t f_t(\mathbf{z})$$

$$\leq \sum_{t=1}^{T} \alpha_t \left( f_t(\mathbf{z}_t) - f_t(\widehat{\mathbf{z}}_{t+1}) - f_{t-1}(\mathbf{z}_t) + f_{t-1}(\widehat{\mathbf{z}}_{t+1}) \right) - \sum_{t=1}^{T} \frac{\lambda(\sum_{i=1}^{t} \alpha_i)}{2} \|\mathbf{z}_t - \widehat{\mathbf{z}}_{t+1}\|^2.$$

**Lemma 5.** *Consider a weighted online learning problem with a series of convex functions $f_1, \ldots, f_T$, and a series of corresponding parameters $\alpha_1, \ldots, \alpha_T$. Let $R(\mathbf{z})$ be a 1-strongly convex regularizer wrt $\|\cdot\|$. The FTRL algorithm, given by*

$$\mathbf{z}_t = \arg\min_{\mathbf{z} \in \mathbb{R}^d} \sum_{i=1}^{t-1} \alpha_i f_i(\mathbf{z}) + R(\mathbf{z}),$$

*achieves the following regret bound:*

$$\forall \mathbf{z} \in \mathbb{R}^d, \quad \sum_{t=1}^{T} \alpha_t f_t(\mathbf{z}_t) - \sum_{t=1}^{T} \alpha_t f_t(\mathbf{z}) \leq R(\mathbf{z}) + 2 \sum_{t=1}^{T} \alpha_t^2 \|\nabla f_t(\mathbf{z}_t)\|_*^2.$$

**Lemma 6.** *Consider a weighted online learning problem with a series of convex functions $f_1, \ldots, f_T$, and a series of corresponding parameters $\alpha_1, \ldots, \alpha_T$. Let $R(\mathbf{z})$ be a 1-strongly convex regularizer wrt $\|\cdot\|$. The FTRL$^+$ algorithm, given by*

$$\mathbf{z}_t = \arg\min_{\mathbf{z} \in \mathbb{R}^d} \eta \sum_{i=1}^{t} \alpha_i f_i(\mathbf{z}) + R(\mathbf{z}),$$

*achieves the following regret bound:*

$$\forall \mathbf{z} \in \mathcal{Z}, \quad \sum_{t=1}^{T} \alpha_t f_t(\mathbf{z}_t) - \sum_{t=1}^{T} \alpha_t f_t(\mathbf{z}) \leq \frac{R(\mathbf{z})}{\eta} - \sum_{t=1}^{T} \frac{1}{2\eta} \|\mathbf{z}_t - \mathbf{z}_{t-1}\|^2. \tag{54}$$

**Lemma 7.** *Consider a weighted online learning problem with a series of convex functions $f_1, \ldots, f_T$, and a series of corresponding parameters $\alpha_1, \ldots, \alpha_T$. Let $\Phi(\mathbf{z})$ be a $\beta$-strongly convex regularizer wrt $\|\cdot\|$. Let $\mathbf{z}_0 \in \mathbb{R}^d$ be the initial point. The OMD algorithm, given by*

$$\nabla \Phi(\mathbf{z}_t) = \nabla \Phi(\mathbf{z}_{t-1}) - \eta \alpha_{t-1} \nabla f_{t-1}(\mathbf{z}_{t-1}),$$

*achieves the following regret bound:*

$$\forall \mathbf{z} \in \mathbb{R}^d, \quad \sum_{t=1}^{T} \alpha_t f_t(\mathbf{z}_t) - \sum_{t=1}^{T} \alpha_t f_t(\mathbf{z}) \leq \frac{D_\Phi(\mathbf{z}; \mathbf{z}_0)}{\eta} + \frac{1}{\beta} \sum_{t=1}^{T} \eta \alpha_t^2 \|\nabla f_t(\mathbf{z}_t)\|_*^2.$$

**Lemma 8.** *Consider a weighted online learning problem with a series of convex functions $f_1, \ldots, f_T$, and a series of corresponding parameters $\alpha_1, \ldots, \alpha_T$. Let $R(\mathbf{z})$ be a 1-strongly convex regularizer wrt $\|\cdot\|$, and $\psi_t(\mathbf{z})$ be the optimism term in round $t$. The Optimistic FTRL algorithm, given by*

$$\mathbf{z}_t = \arg\min_{\mathbf{z} \in \mathbb{R}^d} \sum_{i=1}^{t-1} \alpha_i f_i(\mathbf{z}) + \alpha_t \psi_t(\mathbf{z}) + \frac{1}{\eta} R(\mathbf{z}),$$

*achieves the following regret bound:*

$$\forall \mathbf{z} \in \mathcal{Z}, \quad \sum_{t=1}^{T} \alpha_t f_t(\mathbf{z}_t) - \sum_{t=1}^{T} \alpha_t f_t(\mathbf{z}) \leq \frac{R(\mathbf{z})}{\eta} + \sum_{t=1}^{T} \left[ \frac{\|\alpha_t \nabla f_t(\mathbf{z}_t) - \alpha_t \nabla \psi_t(\mathbf{z}_t)\|_*^2}{2/\eta}. \right] \tag{55}$$

