# OpenReview forum: "Faster Margin Maximization Rates for Generic Optimization Methods"
_NeurIPS.cc/2023/Conference — NeurIPS 2023 spotlight_

### Official Review · Reviewer_Gj8T · 2023-06-21

**Soundness:** 3 good
**Presentation:** 3 good
**Contribution:** 3 good
**Rating:** 7
**Confidence:** 4

**Summary:**

This paper studies efficient algorithms for margin maximization with respect to a general geometry. The problem of margin maximization is interesting because it has been shown that common optimization algorithm such as gradient descent prefers such solutions through a well-known phenomenon called "implicit bias." This paper proposed a novel analytical framework by re-casting the problem as a bilinear game. The main results show that techniques from online learning can be applied to show fast convergence rates for various optimization algorithms such as mirror descent and steepest descent.

**Strengths:**

This paper brings an interesting angle of studying the properties of optimization algorithms through the lens of online learning and games. In particular, Theorem 1 cleanly encapsulates the reduction from optimization to online learning. I feel that this result has a lot of potential in quickly deriving new optimization guarantees by reusing tools from online learning.

I also like the presentation of this paper that it clearly describes prior literature and makes conscious effort in recovering those results through the more general framework of this paper. This allows me to clearly understand the contribution of this paper and appreciate its versatility.

**Weaknesses:**

Notation is quite dense and can be difficult to parse. For example,
1. In section 4.1, it is stated that the mirror descent potential is restricted to $q$-norm for $q \in (1, 2]$. I think it would be better to bring this up sooner in the Preliminaries section.
2. $D_E$ is not a standard notation, why not just call it the KL-divergence?
3. I found that the terms $p$-norms and $q$-norms are used almost interchangeably, which might be confusing to readers.

In Section 4.3, does the mirror descent potential suffer the same limitation as the results in Section 4.1? I have a hard time figuring this out.

The theorem statements are very long and thus look unnecessarily intimidating. In particular, the notion of directional errors is almost entirely ignored in Section 4. I personally would love to see more discussion on the directional error.

The term "rate" is used imprecisely in the introduction. Some of the prior literature measure their rate in terms of margin maximization, while some others in directional error. The author should do more to clarify this.

Regarding the remark on eq. (5)'s computational efficiency. While it is true that its total amount of computation compares similarly to algorithm such as GD, I don't think taking norm of the entire weight vector is practical in a lot of cases, especially for deep learning where parallel or distributed computation is desired.

**Questions:**

I would like to see some proof sketch and strongly recommend the author to add it to the revision. In particular, for Section 4.1, I still don't understand where the $q \in (1, 2]$ limitation comes into play. Does the same limitation apply to Section 4.3?

**Limitations:**

This is purely a theoretical paper, so I don't think this paper has any immediate negative societal impact. I find the proofs to be generally easy to follow, but I did not follow all of them line-by-line.

---

> ### Author Rebuttal · Authors · 2023-08-09
>
> We greatly appreciate your positive and constructive feedback. We will revise the notations and enhance the overall presentation as per your advice. Your specific questions and concerns are addressed below.
>
> >D_E is not a standard notation, why not just call it the KL-divergence?
>
> Thank you for your recommendation. We agree that using standard notation can enhance the readability of our work. In the revised version, we will replace D_E with KL-divergence.
>
> > I would like to see some proof sketch and strongly recommend the author to add it to the revision. In particular, for Section 4.1, I still don't understand where the p∈(1,2] limitation comes into play.
>
> We appreciate your interest in the derivation details, and we agree that a proof sketch would provide valuable insights. In the revised version, we will include a more comprehensive discussion in Section 4.1 to explain why the value of q must fall within the range (1,2]. The key to understanding this is that when q is in the range (1,2], it ensures the (q-1)-strong convexity of $\ell_t(w)$, which in turn ensures a sufficiently small regret bound for the w-player. It's important to note that, as observed in Theorem 7 (Appendix C), the p-player has a significantly larger (potentially linear) regret bound. Therefore, it is crucial for the w-player's regret bound to be negative and small enough to offset terms in the final bound.
>
> >In Section 4.3, does the mirror descent potential suffer the same limitation as the results in Section 4.1?/ Does the same limitation apply to Section 4.3?
>
> Indeed, the same limitation applies to the results presented in Section 4.3. As outlined in Theorem 6 (Section 4.3), we assume the norm $||\cdot||$ to be such that $||\cdot||^2$ is strongly convex. Although this is slightly more general, it essentially confines us to cases where q falls within the range (1,2]. We will make sure to emphasize this point in the revised version for clarity.

---

> > ### Comment · Reviewer_Gj8T · 2023-08-10
> >
> > Thanks for your response. It would nice if you offer a preview of the additional discussion you intend to add in the revision. I do not feel comfortable with moving my score before then.

---

> > > ### Author Response · Authors · 2023-08-11
> > > **Response**
> > >
> > > Thank you very much for your prompt feedback! The following is a preview of the discussion we would like to add:
> > >
> > >
> > >
> > > **Regarding the range of q**: In the original paper, we discussed the basic idea of why sublinear regret can be achieved (Lines 216-224):
> > >
> > > > This is an interesting and unusual design because the regularized greedy algorithm will clearly suffer a worst-case linear regret for the p-player. Luckily, we find that for our specific problem, the dominating term of the p-player’s regret bound can be canceled by the w-player’s regret bound, which is negative as the corresponding algorithm used is clairvoyant, i.e. can see the current loss `t before making a decision at round t. This ensures that sublinear (and more generally fast) rates are possible.
> > >
> > > In the revised version, we intend to add a remark below this paragraph and be clear about the role of strong convexity. Specifically:
> > >
> > > To be more concrete, for the p-player, we show it guarantees a data-dependent regret bound:
> > > $$
> > > Reg_T^p = O\left(\sum_{t=2}^T\frac{(t-1)(q-1)}{2}||w_t-w_{t-1}||_q^2 + \log n\log T\right),
> > > $$
> > >
> > > which can be as worse as $O(T)$. On the other hand, for the w-player, the regret is bounded by
> > > $$
> > > Reg_T^w = O\left(-\sum_{t=2}^T\frac{(t-1)(q-1)||w_t-w_{t-1}||_q^2}{2}\right)
> > > $$
> > >
> > >
> > >
> > > which cancels the leading term in $Reg_T^p$ and leads to a small $C_T$. Note that, FTL$^+$ (w-player's online algorithm) can only achieve such a bound when q\in(1,2], as in this case $\ell_t(w)$ is $(q-1)$-strongly convex. If $q>2$, the strong convexity not longer exists. In this case, FTL$^+$ only ensures a zero regret bound, which is insufficient to achieve a sub-linear C_T.
> > >
> > >
> > >
> > > **Regarding Section 4.3**: Following your question, we will add the following discussion at Line 299: Finally, we note that, Theorem 6 requires $||\cdot\||^2$ to be strongly convex, which is satisfied when for q-norm with $q\in(1,2]$.
> > >
> > > **Regarding the directional error**: In the original paper, we removed the conclusion on the directional error due to lack of space. In the final version (with more spece), we will add them back to the main theorems. One preview can be found in Theorem 7 of Appendix C, which contains the bounds on the margin and directional error.
> > >
> > >
> > >
> > > **Regarding the computational complexity**: Following your suggestion, we will add the following after Line 207: we note that, the p-MD algorithm of Sun et al., (2022) does not need to compute the norm of the decision at each round, which can be more efficient in real-world  applications where parallel or distributed computation is desired.

---

> > > > ### Comment · Reviewer_Gj8T · 2023-08-11
> > > >
> > > > Thank you for your prompt responses. When you are talking about strong convexity, do you mean strong convexity wrt to the $\ell_2$ norm or the corresponding $\ell_q$ norm?

---

> > > > > ### Author Response · Authors · 2023-08-11
> > > > > **Response**
> > > > >
> > > > > Thank you for the discussion.
> > > > >
> > > > > For Section 4.1, we mean for $q\in(1,2]$, the function $\frac{1}{2}||\cdot||_q^2$ is (q-1)-strongly convex with respect to the $\ell_q$ norm (it can be found in, e.g., Lemma 6.19 of https://arxiv.org/abs/1912.13213). Therefore, $\ell_t(w)$ is also  (q-1)-strongly convex with respect to the $\ell_q$ norm.
> > > > >
> > > > > For Section 4.3, we assume the function $\frac{1}{2}||\cdot||^2$ is $\lambda$-strongly convex with respect to the corresponding $||\cdot||$ norm. Therefore, $q\in(1,2]$ satisfies this assumption.
> > > > >
> > > > > Thank you!

---

> > > > > > ### Comment · Reviewer_Gj8T · 2023-08-11
> > > > > >
> > > > > > Thank you for thoroughly answer all of my questions. I will raise my score to **7**.

---

> > > > > > > ### Author Response · Authors · 2023-08-11
> > > > > > > **Response**
> > > > > > >
> > > > > > > Thank you very much for the constructive discussion and positive evaluation!

---

### Official Review · Reviewer_VcXn · 2023-07-06

**Soundness:** 3 good
**Presentation:** 3 good
**Contribution:** 2 fair
**Rating:** 6
**Confidence:** 2

**Summary:**

This work gives a unified perspective on maximal margin problems realized by a wide range of optimization algorithms. It mainly covers three cases: (i) steepest descent under a general norm, (ii) mirror descent, and (iii) momentum-based acceleration. The authors collectively refer to them as generic optimization problems. The essential point of the theory is that they translate these generic optimization problems into a regularized bilinear game with online learning. This enables us to derive relatively tight margin maximization rates and to characterize the implicit bias of each algorithm.

**Strengths:**

This work is based on a concrete and rigorous theoretical analysis of a wide range of optimization problems. One interesting point is that all of them can be mapped in the regularized bilinear game formulation. It gives us a unified perspective of various optimization problems. In addition, this unified formulation enables us to obtain tight convergence rates compared to the previous work.

**Weaknesses:**

While this work covers various optimization problems and an interesting formulation with the bilinear game, it might be possible to view this as a collection of results that are not particularly novel. The convergence rates of the problems (for example, (i) steepest descent under a general norm, (ii) mirror descent, and (iii) momentum-based acceleration) have been studied although they seem looser than those obtained in the current work. The game formulation has been also provided by some previous work ([Wang et al 2021, 2022b]) in a limited situation.


**Questions:**

**Difference from previous work**

Since this work covers various topics of optimization problems, it seems hard for beginners in this research area to judge at what points it is significantly novel compared to the previous work. In particular, the difference from Wang et al 2022b seems to need more clarification. Did this previous work not address the max-margin rate of all of (i) steepest descent under a general norm, (ii) mirror descent, and (iii) momentum-based acceleration problems?

**Exetention to other generic loss functions**

It is quite curious whether the idea of using the bilinear game can be extended to other loss functions except for the exponential loss. Is there any work or evidence that the current proof approach could potentially be applied to other loss functions? Otherwise, is it quite specific to the exponential function?

**Tightness compared to numerical experiments**

Although I agree that the theoretical evaluation of the bounds itself contributes to enriching our understanding of the problems, it remains unclear how tight the obtained convergence rate is.  I mean, there would be the possibility that the obtained bounds could be much loose compared to the real optimization in experiments. The authors show no empirical confirmation and this makes a bit unclear the superiority of the fast convergence rates obtained in this work. It is not a major flaw, though.

**Limitations:**

As I mentioned in Questions, one limitation is that there is no comparison with numerical experiments and tightness (or "true" implicit bias) seems unclear. The other is the exponential loss as is mentioned in Section 5.

---

> ### Author Rebuttal · Authors · 2023-08-09
>
> > Difference from Wang et al 2022b seems to need more clarification. Did this previous work not address the max-margin rate of all of (i) steepest descent under a general norm, (ii) mirror descent, and (iii) momentum-based acceleration problems?
>
> Thank you for the constructive suggestion; we will add a more detailed discussion in the revised version. Compared with Wang et al (2022b), we note that:
>
> 1)   Wang et al., (2022b) draws the connection between Nesterov-accelerated GD for ERM and solving the bilinear game through an online dynamic. However, it was unclear whether this kind of analysis suits other gradient-descent-based methods, and generic optimization methods such as mirror descent/steepest descent was not addressed at all. We observe in this work that the non-linearity of the mirror map in generic optimization methods, such as mirror descent and steepest descent, makes the analysis particularly challenging. In this paper, we reveal that the game framework can in fact encompass implicit bias analysis for a range of generic optimization methods, and offer a more streamlined and unified analysis.
> 2)   Wang et al., (2022b) also proposed an accelerated p-norm perceptron problem. However, they only demonstrated that the algorithm could achieve a non-negative margin, leaving open questions regarding whether the margin can be maximized (i.e., converge to $\gamma$), and if so, what the margin maximization rate would be; 2) They only presented the online dynamic, without its equivalent optimization form under ERM.
>
> > It is quite curious whether the idea of using the bilinear game can be extended to other loss functions except for the exponential loss. Is there any work or evidence that the current proof approach could potentially be applied to other loss functions? Otherwise, is it quite specific to the exponential function?
>
> Thank you for this intriguing question, and we will add a more detailed discussion in the revised version. please refer to the first question in the general response for our answer to this question.
>
> > Tightness compared to numerical experiments
>
> We appreciate your valuable suggestions. While our paper's primary focus lies in the theoretical aspects of implicit bias analysis, we concur that supplementing these theories with numerical experiments would enhance our work. Accordingly, we will introduce experiments in the revised version to substantiate the efficacy of our proposed methods.

---

> > ### Comment · Reviewer_VcXn · 2023-08-14
> >
> > Thank you for your kind response and clarification.
> >
> > >we will add a more detailed discussion in the revised version. Compared with Wang et al (2022b), ...
> >
> > >Accordingly, we will introduce experiments in the revised version to substantiate the efficacy of our proposed methods.
> >
> > I am looking forward to seeing them in the final version.
> > I guess that even very brief experiments will become informative for subsequent works.
> >
> > Since I understand that the lack of such empirical observation does not flaw the main contribution, I feel comfortable keeping my score on the accept side.

---

### Official Review · Reviewer_bPGG · 2023-07-15

**Soundness:** 4 excellent
**Presentation:** 3 good
**Contribution:** 4 excellent
**Rating:** 8
**Confidence:** 2

**Summary:**

The paper studies the implicit bias of generic optimization methods, which plays a key role in understanding their generalization capabilities in settings with multiple solutions.

The authors propose a new game framework to derive margin and directional error rates, which consists of transforming the optimization method into an equivalent instance of a 2-player margin-maximization game. Afterwards, margin and directional error rates can be derived directly from the players’ average online regret.

They show equivalent transformations (under exp loss) from weighted MD with a squared p-norm potential and SD under a general norm to instances of the proposed 2-player game, yielding convergence rates from each transformation’s induced regret. More aggressive learning rates are also studied and shown to be able to improve convergence rates.

The paper shows, through their game framework, that even faster rates can be achieved with Nesterov MD and a form of momentum SD.

**Strengths:**

The paper is clear and well-written, adopting consistent, well-defined notation / objects and presenting assumptions, definitions, and results in a local and organized fashion.

Its novelty and contribution seem significant:

The proposed game framework seems to be a novel approach to study implicit biases (margin / directional error rates). By reducing rate analyses to the task of finding equivalent transformations to the 2-player game, it might facilitate the study of new method’s implicit biases and hence prove useful for future research.

It provides a fresh perspective and, as shown in the paper, accommodates different forms of MD and SD, suggesting that it might be able to capture different first-order methods as well. The transformations, although technical and non-trivial, provide valuable insights on the behavior of MD and SD.

The use of clairvoyance for player w to show form equivalence is also interesting and might be useful for new proof techniques, but I cannot assess its novelty since I am not very familiar with the related work.

The improved margin rates and new accelerated methods given in Section 4.3 are also impactful and suggest that momentum might be crucial to speed up (general) margin maximization – from what I understand the community only had evidence of this for $p=2$ (Ji et al’21).

From my understanding, the paper shows the first $1/T^2$ margin rate for $p \in (1,2)$, which seems to be a significant contribution.


**Weaknesses:**

The last page or two seem to have been written a bit in a hurry, and little space has been allocated to Section 4.3 compared to its significance. However, since these can be easily fixed in a revision I will not account for that when rating the paper.

Although I believe the paper has enough contributions, it would be interesting to have some small (even if synthetic) experiments with margin by time curves for different methods and learning rates regimes.

Minor fixes:

L51: log t -> log T?

L{276, 285, L291, L293, L295}: Algorithm 4.3 -> Algorithm 4

L516 proof -> prove

Should state in Section 4.3 that proof of Theorem 6 is in Appendix E

Eq 6 and, Eq 7: period instead of comma after equation

Eq 8: missing period after equation

Equation at end of page 6: colon instead of period before eq, period after eq.


**Questions:**

Is the potential in the left box of Algorithm 4 the general norm squared? This seems to be the case but I think it is not specified in Section 4.3.

**Limitations:**

Limitations are properly acknowledged, e.g. applicability of the framework only to exp loss and showing method-game equivalence might take effort and be non-trivial.

---

> ### Author Rebuttal · Authors · 2023-08-09
>
> We deeply appreciate your constructive review and supportive comments on our work! We will refine the presentation and rectify the typos in line with your suggestions. For clarity, we will specify in the revised version that the potential in the left box of Algorithm 4 refers to the general norm squared.

---

> > ### Comment · Reviewer_bPGG · 2023-08-16
> >
> > Thanks for the response!

---

### Official Review · Reviewer_HfJS · 2023-07-16

**Soundness:** 3 good
**Presentation:** 3 good
**Contribution:** 3 good
**Rating:** 6
**Confidence:** 2

**Summary:**

This paper studies the implicit bias of the generic optimization method such as mirror descent and steepest descent. By transforming the generic optimization algorithm into an online learning dynamic, this paper shows the accelerated rate and offers a new perspective.

**Strengths:**

1. This paper is well-written.
2. The theoretical analysis is solid and explicitly explained

**Weaknesses:**

The major weakness is that this paper only focuses on exponential loss.
In addition, choosing the hyperparameters is sometimes difficult.

**Questions:**

1. Can the author provide some simple experiments to verify the correctness of the theoretical results? For example, using the synthetic dataset with its max-margin solution known.
2. For the P play, will solving the subproblem lead to a larger complexity?
3. What is the meaning of $D_E$ in Algorithm 3?


Typos:
Algorithm 4.3 should be Algorithm 4.

**Limitations:**

OK

---

> ### Author Rebuttal · Authors · 2023-08-09
>
> Thank you very much for the review and positive comments!
>
> > Can the author provide some simple experiments to verify the correctness of the theoretical results? For example, using the synthetic dataset with its max-margin solution known.
>
> Thank you for the constructive suggestion; as we discuss in the second point of the general response, we will add numerical experiments in the revised version.
>
>
> > For the P play, will solving the subproblem lead to a larger complexity?
>
> Thank you for the question. We note that the online/game framework is only introduced for theoretical analysis, and we do not need to actually run the corresponding online algorithms. Therefore, the p-player will not introduce extra computational complexity. We will make this clearer in the revised version.
>
>
> > What is the meaning of D_E in Algorithm 3?
>
> Thank you for the question. D_E stands for the Bregman divergence with respect to the negative entropy regularizer. We will make it clear in the notation part.

---

### Official Review · Reviewer_bnuX · 2023-07-17

**Soundness:** 4 excellent
**Presentation:** 3 good
**Contribution:** 3 good
**Rating:** 7
**Confidence:** 2

**Summary:**

This work introduces a novel method to derive margin maximization and directional error rates for generic optimization methods. The method consists in finding a reformulation of the regular optimization as a minmax bilinear game. Rates can then be derived using online learning techniques. The authors demonstrate (Thm. 1) how solving a min-max regularized bilinear objective using online learning is maximizing the margin. They also derive (Thm. 2,3,5) the minmax formulation of mirror descent and steepest descent. Using their method, several new margin-maximization and directional error rates for mirror descent (average iterate) and steepest descent are derived. Moreover, rates are also derived for accelerated methods (Thm. 4 and 6).

**Strengths:**

Originality & Significance: While not being entirely familiar with all the prior works, I find the main idea presented in this paper novel. The proposed rates are improving upon prior works and the approach seems to be able encompass a wide range of optimization techniques, making it a significant contribution in my opinion.

Clarity: I found the paper easy to read despite its density. Some small typos: in Figure 1, this should be max min and not max max.

**Weaknesses:**

The authors ask the question whether generic optimization methods can achieve faster rates than GD. It seems the proposed rate for steepest descent is matching the GD rate, but the question is still open for mirror-descent. It is unclear how to conclude for mirror descent as you are deriving an average iterate rate that is difficult to compare to existing last iterate rates. Usually average iterate rates are better than last iterate rates.

It is unclear how good those rates can become. Is there a lower-bound you could comment on?

This work provide a unifying method to obtain margin maximization rates for linear models with exponential loss. It is unclear how the approach could be extended to different losses.

**Questions:**

What would it take to remove the $\log(T)$ term in the $\mathcal{O}(\frac{\log n \log T}{(q-1) T})$ for mirror descent?
For mirror descent, what is preventing the analysis to be extended to the last iterate?

What would a comparison between your mirror descent rate and existing GD rates look like in the average iterate regime?

Do you believe a similar approach could be used beyond the exponential loss?

**Limitations:**

Limitations have been discussed in the conclusion.

---

> ### Author Rebuttal · Authors · 2023-08-09
>
> We are profoundly grateful for your positive evaluation of our work and your detailed, constructive feedback. We address some of your specific inquiries below:
>
> >What would be required to eliminate the term $O(\log T)$ in the $O(\frac{\log n\log T}{(q-1)T})$ for mirror descent?
>
> We appreciate this insightful question. Following the proof structure of Corollary 3, if we fix the total number of iterations $T$ in advance, we can set $\alpha_t$ to $T$, thereby eliminating the $\log T$ term. However, as it does not make sense to fix $T$ in an optimization application of online learning, we set $\alpha_t$ to either $1$ or $t$, both of which introduce a $\sum_{t=1}^T \frac{1}{t}$ term, resulting in the $\log T$ factor. We will elaborate on this point in our revised version.
>
> >Regarding mirror descent, what hinders the analysis from being extended to the last iterate?
>
> This is an excellent question. At present, the primary challenge lies in the non-linearity of the mirror map, namely $\nabla \Phi(\sum_{t=1}^Tw_t)\not=\sum_{t=1}^T \nabla\Phi(w_t)$. While the equality holds for $q=2$, it does not when $q\in(1,2)$, posing difficulties for various aspects of the proof such as obtaining a lower bound on the weighted sum of $w_t$ and proving algorithm equivalence. We discovered that by allowing the p-player to implement a specialized algorithm (namely, regularized greedy as defined at the bottom of page 6), we can derive a weighted average version of MD via algorithm equivalence analysis, and the issues mentioned earlier can also be addressed in a nuanced manner. We will provide a more thorough discussion on this issue.
>
> > How would a comparison between your mirror descent rate and the existing GD rates appear in the average iterate regime?
>
> Thank you for the question. When $q=2$, our average MD corresponds to an average version of GD and exhibits an $O(1/T)$ rate, which is similar to the optimal rate of last-iterate GD. Conversely, when $q=2$, our last-iterate steepest descent algorithm also simplifies to last-iterate GD, thereby demonstrating that our framework is versatile and can be used to analyze last-iterate GD.
>
> > Do you believe a similar approach could be used beyond the exponential loss?
>
> Thank you for raising this point. Please refer to the first question of the general response for our answer to this inquiry.

---

> > ### Comment · Reviewer_bnuX · 2023-08-15
> >
> > Thank you those clarifications!

---

### Author Rebuttal · Authors · 2023-08-09

We are deeply grateful to all reviewers for their positive feedback and valuable suggestions. We commit to implementing your advice to refine our paper, and some of the frequently asked questions are addressed below.

>Do you believe a similar approach could be used beyond the exponential loss?

We appreciate this insightful question and will include a more comprehensive discussion on this topic in our revised version. Our primary aim in this paper is to present a unified analysis framework for understanding the implicit bias phenomenon and to provide faster convergence rates for a variety of generic optimization methods. We posit that the analysis of exponential loss serves as a first step, and the application of the game/online learning framework bears the potential to be broadened to incorporate an analysis of a more diverse array of loss functions. The current choice of exponential loss is particularly apt for the game/online dynamic analysis as it relates closely to the classical Hedge algorithm in online learning, which also employs the exponential function to measure experts' losses. To broaden our scope to more general functions, we could contemplate replacing the exponential loss in the Hedge algorithm with other losses (e.g., using the “Polynomially Weighted Average Forecaster”, given in Corollary 2.1 of Cesa-Bianchi & Lugosi (2006)), and attempt to establish a link between this online dynamic and optimization methods for more general loss functions. Our preliminary investigations suggest that this avenue is promising but highly non-trivial, and warrants further exploration in future work.

> Tightness measured through numerical experiments in addition to theory.

We value your constructive suggestions. While our paper primarily focuses on the theoretical aspects of implicit bias analysis, we concur that complementing our theoretical findings with numerical experiments would provide a more robust validation of our work. Consequently, we intend to incorporate experiments in the revised version to demonstrate the effectiveness of the proposed methods.


Cesa-Bianchi, N., & Lugosi, G. Prediction, learning, and games. Cambridge university press 2006.

---

### Decision · Program_Chairs · 2023-09-21

**Decision:**

Accept (spotlight)

**Comment:**

This very nice paper generalizes the (dual) momentum-based method for margin maximization of Ji-Srebro-Telgarsky, leading to an adjusted proof technique with new insights, and $1/T^2$ rates in broader settings, and in particular for other norms. The paper is dense with many results and the authors will have a tough time fitting it in a poster, let alone a spotlight! Nice work.

Minor comments:

- The camera ready gets an extra page; I urge the authors to use that space to import more insights from the appendix.
- As this work has some focus on margins wrt different norms, I recommend mentioning the 2013 work "margins, shrinkage, and boosting" by Telgarsky which not only gives a  $1/\sqrt{T}$ rate in  $\\|\cdot\\|_1$ (though it is presented there as a $1/T$ rate to a suboptimal margin, but moreover this work gave the ln-sum-exp proof idea as well as the large step size idea used in the later cited works by various authors (and is cited by them for this).
- Seems the *very* last period in the appendix should be outside the parens.
- The huge equation block in the appendix (currently marked line 482) is what causes whitespace issues elsewhere in the appendix; please break it up to aid other appendix readability/presentation.